# Agroinfiltration Mediated Scalable Transient Gene Expression in Genome Edited Crop Plants

**DOI:** 10.3390/ijms221910882

**Published:** 2021-10-08

**Authors:** Maninder Kaur, Pooja Manchanda, Anu Kalia, Farah K. Ahmed, Eugenie Nepovimova, Kamil Kuca, Kamel A. Abd-Elsalam

**Affiliations:** 1School of Agricultural Biotechnology, College of Agriculture, Punjab Agricultural University, Ludhiana, Punjab 141004, India; maninderkaur-coasab@pau.edu; 2Electron Microscopy and Nanoscience Laboratory, Department of Soil Science, College of Agriculture, Punjab Agricultural University, Ludhiana, Punjab 141004, India; kaliaanu@pau.edu; 3Biotechnology English Program, Faculty of Agriculture, Cairo University, Giza 12613, Egypt; farahkamel777@gmail.com; 4Department of Chemistry, Faculty of Science, University of Hradec Kralove, 50003 Hradec Kralove, Czech Republic; eugenie.nepovimova@uhk.cz; 5Biomedical Research Center, University Hospital Hradec Kralove, 50005 Hradec Kralove, Czech Republic; 6Plant Pathology Research Institute, Agricultural Research Center (ARC), 9-Gamaa St., Giza 12619, Egypt; kamelabdelsalam@gmail.com

**Keywords:** *Agrobacterium*, CRISPR/Cas9, genome editing, targeted site modification, transgene-free, transfer-DNA

## Abstract

*Agrobacterium*-mediated transformation is one of the most commonly used genetic transformation method that involves transfer of foreign genes into target plants. Agroinfiltration, an *Agrobacterium*-based transient approach and the breakthrough discovery of CRISPR/Cas9 holds trending stature to perform targeted and efficient genome editing (GE). The predominant feature of agroinfiltration is the abolishment of Transfer-DNA (T-DNA) integration event to ensure fewer biosafety and regulatory issues besides showcasing the capability to perform transcription and translation efficiently, hence providing a large picture through pilot-scale experiment via transient approach. The direct delivery of recombinant agrobacteria through this approach carrying CRISPR/Cas cassette to knockout the expression of the target gene in the intercellular tissue spaces by physical or vacuum infiltration can simplify the targeted site modification. This review aims to provide information on *Agrobacterium*-mediated transformation and implementation of agroinfiltration with GE to widen the horizon of targeted genome editing before a stable genome editing approach. This will ease the screening of numerous functions of genes in different plant species with wider applicability in future.

## 1. Introduction

The CRISPR-Cas9 (Clustered Regularly Interspaced Short Palindromic repeat DNA Sequences- CRISPR associated), a Nobel prize-winning technology for the year 2020 pioneered by Emmanuelle Charpentier and Jennifer Doudna in 2012 proved to be an indispensable tool for editing genomes in the agricultural biotechnology field. Besides genome editing, this tool has been applied for studying gene regulation, epigenetic editing and chromatin engineering also [1]. The discoveries of meganucleases, zinc finger nucleases (ZFNs), Transcription Activator-Like Effector Nucleases (TALENs) followed by the breakthrough discovery of CRISPR-Cas9 have revolutionized the genome editing research. It involves targeted base editing of the genome through the use of *Streptococcus pyogenes* Cas9 endonuclease and short guide RNA molecule (sgRNA) that revolutionized genetics and functional genomics [2,3]. The Cas9 endonuclease and sgRNA lead to the generation of DNA double-stranded breaks in the targeted genome sequences and ensure successful editing of plant genomes [4].

Plant biotechnology which involves the generation of genetic modifications of host cells through the introduction of foreign genes encased in a vector leads to the production of transgenic plants. These genetic modifications can be carried out through numerous methods namely indirect gene transfer methods like *Agrobacterium*-mediated, floral dip, agroinfiltration, and direct gene transfer methods like polyethylene glycol method (PEG)-mediated method, particle bombardment (biolistics), microinjection, and electroporation (Figure 1). These serve as a key for the development of genetically engineered plants with the trait of interest in a limited time that circumvents the conventional plant breeding approaches. In all the above-mentioned methods, widely acceptable genetic transformation method includes indirect transfer using *Agrobacterium* [5]. It is one of the most preferred techniques for the generation of genetically modified plants [6] as *Agrobacterium*-a natural genetic engineer, can deliver the desired gene of interest naturally due to its innate ability to transfer T-DNA. Scientists have explored their vision for using the genome editing process in conjugation with the *Agrobacterium*-mediated process through the use of Cas9 and sgRNA in the T-DNA to perform specific base editing. 

Recently, Sandhya et al. [14] reviewed numerous CRISPR/Cas9 based delivery methods such as *Agrobacterium*-mediated, biolistics, PEG-mediated, and floral dip or pollen-tube pathway method. But all these methods require sophisticated strategies for the selection of desired transformants possessing novel characteristics. Moreover, numerous confirmation protocols are required to ascertain the successful transfer of the gene and proper functioning of the gene of interest in the host plant [15]. The *Agrobacterium*-based floral dip method undergoes a stable transformation that targets the germinal cells while *Agrobacterium*-based transient approach i.e., agroinfiltration target somatic cells, hence not transferred to the next generation. The genetic engineering technology mostly suffers from major limitations like low transformation efficiency, high time requirements, and event of insertion of foreign DNA in the host plant leading to stable transformation [16].

Keeping in view the generation of stable transformation through *Agrobacterium*-based technique that involves tissue culture, we simply focus on *Agrobacterium*-based transient agroinfiltration approach which does not involve time-consuming tissue culturing step [17]. As the CRISPR/Cas9 DNA editing still has regulatory and biosafety concerns [18,19], therefore, transient expression through agroinfiltration-based genome editing holds promise in the present era that will overcome the legislation concerns due to its temporary expression for a limited time ~within 3 days [20].

## 2. Genome Editing Methods

The CRISPR/Cas9 system has been successfully applied in various plant species for site-specific base modifications through various genetic transformation methods with varying editing efficiency. It has been observed that the editing efficiency of 9.6%, 18.4%, and 31.9% can be obtained in the edited lines depending on the *Agrobacterium*-mediated transformation, ribonucleoprotein complexes, and transient expression in protoplasts respectively [21]. Another advancement of the CRISPR/Cas9 genome editing, the Transiently Expressed CRISPR/Cas DNA (TECCDNA) is a simple and efficient genome-editing approach in which mutant plants are regenerated after transient expression of CRISPR/Cas9 DNA [22]. The CRISPR/Cas9 DNA (plasmid constructs) or RNA (in vitro synthesized transcripts) can be delivered through biolistic technique. The circumvention of lengthy and labour-intensive herbicide selection steps on a medium supplemented with antibiotic was escaped concluding that the foreign DNA does not get integrated into the genome. Generally, the plant genome editing tools require additional cycles of plant regeneration under antibiotic selection medium but TECCDNA approach excluded the addition of herbicide or antibiotics addition to the medium meant for the selection of transformed plants. After biolistic delivery of TECCDNA, embryos were transferred to callus induction, regeneration, and rooting medium that generated a large number of T_0_ seedlings about 1 week later provided no selective agents were used during the tissue culture process. After TECCDNA, CRISPR-Cas ribonucleoprotein (RNP) served as a simple, convincing, and promising tool for precision plant breeding [23] (Figure 2). This technique has been used for targeted genomic modifications without causing any genomic disturbances or off-target effects [24]. Contrastingly, agroinfiltration involves the direct introduction of a foreign gene in the intercellular [7] and extracellular [25] leaf spaces thereby circumventing the requirement of expensive biolistic equipment for both TECCDNA and CRISPR-Cas RNP approaches. The use of lengthy procedures and sophisticated/ specialized equipment can be avoided through the implementation of the agroinfiltration approach. 

Specific reviews [29,30,31,32,33] are available on transient gene expression in different host organisms. Sheludko [29] reviewed the biological characteristics of the transient expression process and application for recombinant protein production. Tyurin et al. [33] also reviewed the main and critical steps involved in various methods for transient gene expression in plants. Another recent review by Zlobin et al. [4] elaborated the utilization of *Agrobacterium*-based floral dip transformation for genome editing. However, the floral dip transformation method falls short for targeting a single gene due to several specific problems such as non-reproducibility of the process, generation of false-positive results, and low transformation efficiency. Therefore, the agroinfiltration transient approach has several applications in genome editing which have numerous advantages over stable transformation. The primary objective of this compilation is to pursue the plant researcher’s focus on the transient approach of agroinfiltration-based genome editing for site-specific modification research. Therefore, the first comprehensive compilation on the use of agroinfiltration in CRISPR-Cas9 mediated genome editing and implementation of agroinfiltration-based genome editing method(s) in plants.

## 3. Priority towards *Agrobacterium*-Based Genetic Transformation

*Agrobacterium tumefaciens* holds a prominent stature in plant biotechnology since its discovery in the late 18th century till date. Since, the first genetically modified plants using *Agrobacterium*-mediated genetic transformation process in the year 1983, plant genetic transformation has become an indispensable tool for plant biology, crop improvement and commercial farming interventions [34]. It generally refers to the process of alteration of genetic constituents in the host plant through the introduction of the gene of interest through genetic engineering to achieve desired gene expression [35]. *Agrobacterium* harboring the tumor-inducing (Ti) plasmid although responsible for causing crown gall, cane gall, and hairy root disease yet it has become the most popular plant transformation tool that has the requirement of two genetic components i.e T-DNA and virulence region located on the bacterial Ti-plasmid. The plant researchers developed the recombinant *Agrobacterium* strains that do not lead to tumor formation but possess the capability to transfer gene(s) of interest to pursue plant transformation [36]. Its success is based on numerous genetic transformation methods that exhibit few advantages and disadvantages. The biggest advantage includes the utilization of genetic transformation methods with the advent of genetic engineering which evades time-consuming techniques for the transfer of desired traits through traditional plant breeding techniques. 

The indirect gene transfer involves the introduction of foreign DNA in the host plant utilizing biological vectors (*Agrobacterium*). While direct gene transfer techniques involve direct delivery of foreign DNA in the plant genome through various physical or chemical agents such as particle bombardment [8,37], microinjection [38], PEGylation or calcium phosphate mediated protoplast transformation [39,40], electroporation [41,42], and silicon carbide whisker-enabled transformation [43] (Figure 1).

The major advantage of the *Agrobacterium*-mediated transformation process is a low-cost transfer of a single copy of the gene of interest omitting out the problem of gene silencing [44,45,46,47,48,49,50,51]. Hwang et al. [52] reviewed the advantages of the *Agrobacterium* method comprehensively. They have presented it as the most frequently used and most popular method of genetic engineering in the present era. Also, its considerable and ever-expanding contribution for the elucidation of fundamental mechanisms to study gene regulation or protein function in transgenic plants in basic biology research is noteworthy [53].

The indirect transfer of foreign DNA through *A. tumifaciens* utilizes the innate ability of *Agrobacterium* to transfer intact T-DNA through Type 4 secretion system [52,54,55]. It favours trans-kingdom DNA transfer [56]. 

Earlier, *Agrobacterium*-mediated transformation process was possible only in dicots due to the release of acetosyringone (phenolic compound) but nowadays, this method can be applied broadly in the cereal species, gymnosperms, yeast, and many filamentous fungi [57,58]. *A. tumefaciens* can transform non-host also including fungi, algae, sea urchin embryos, human cells, and the gram-positive bacterium *Streptomyces lividans* [59,60,61,62,63,64,65,66,67,68,69,70]. The wide host range of *Agrobacterium* equips it to be effectively utilized for the generation of transgenic plants [71]. *Agrobacterium*-mediated genetic transformation is applicable to plant species, variants, and cultivars for plant genome modifications [72,73,74]. Moreover, *A. tumefaciens* can be equally utilized for both transient and stable transformation methods in plants. 

Keeping in view the advantage of this phytopathogen, the complex *Agrobacterium*-plant interactions have been reviewed extensively [31,75,76,77,78,79,80]. Recently, Nonaka et al. [81] have developed the Super-*Agrobacterium* ver. 4 that has major application in basic plant science research. It has been reported that Super-*Agrobacterium* ver. 3 was effective in the agro-infiltration method [82,83].

## 4. Types of *Agrobacterium*-Based Genetic Transformation

The *Agrobacterium*-based genetic transformation undergo two specific mechanisms; stable (T-DNA integration in the genome stably inherited) [84,85,86,87] and transient transformation (absence of T-DNA integration in the genome still capable to perform transcription and translation) [52,88,89,90,91,92,93,94]. The expression of T-DNA carrying the gene of interest occur in either transient or stable manner was demonstrated by Krenek et al. [30], Janssen and Gardner [95]. Both stable and transient transformation methods can be utilized in genome editing [96], to study the gene and protein function [97,98], molecular processes [99] and are applicable to numerous plant species [100]. Heenatigala et al. [99] have developed efficient protocols for stable and transient gene transformation for *Wolffia globosa* using *Agrobacterium* that has application in a wide range of scientific and commercial processes. Stable genetic transformation process is more appropriate to study long-term analysis of gene function or for long-term production of specific compounds but has the requirement of several months. However, if the target is to introduce a gene of interest in a short period to produce desired protein products, the preferred method is transient transformation [99]. Hence, transient transformation is a preferred method to study short-term gene expression analysis [101]. 

## 5. Stable Genetic Transformation

The transfer of T-DNA through *Agrobacterium* leading to the inheritance in the next generation referred as stable transformation. Generally, this method confers the gene transfer through *Agrobacterium* via two means; co-cultivation method (including tissue culture technology) and floral dip method (devoid of tissue culture). The co-cultivation method using *Agrobacterium* suspension containing desired gene of interest involves culturing the explant. Although it leads to the formation of stable transformants yet it suffers from several disadvantages such as non-uniform transformants can be obtained due to genetic and epigenetic changes that occur during repeated culturing. The induction of callus during tissue culture is a cumbersome approach leading to the generation of undesirable somaclonal variations thus regeneration protocols have not been developed for all plant species. 

The widely adapted *Agrobacterium*-mediated stable gene transfer method is the floral dip method. The floral dip method avoids the subsequent tissue culturing steps that may lead to generation of somaclonal variants such that desired transformants can be obtained within a short time period as compared to the co-cultivation method [52]. 

## 6. Transient (Temporary) Genetic Transformation

The temporary integration of the exogenous DNA in the plant progeny is the basis of transient transformation of plant tissues [102]. *Agrobacterium*-mediated genetic transformation involves transfection of millions of plant cells by T-DNA in a short period. It allows only a short-term (transient) expression of the gene constructs in a plant cell. The transient approach holds the fact that disarmed *Agrobacterium* strains lead to modification of non-reproductive or somatic tissue which is not inherited and hence transient in nature [103]. It has been reported that heritable mutations adhering to Mendel’s law only have the potential to transfer to the next generations [104] but this is not the case in agroinfiltration because transgenes are transiently expressed in somatic cells of plant tissues and not germ cells hence not heritable. Transient gene expression has been demonstrated to produce large amounts of recombinant proteins in a very short time of few days [105,106]. The somatic cells in the zone of agroinfiltration have the potential to express the transgenes under the control of the constitutive CaMV-35S promoter within 3–5 days after agroinfiltration. A transient expression is an approach for the verification of transformation construct activity and the validation of the formation of small amounts of recombinant protein. Potrykus [107] reviewed the most widely used *Agrobacterium* infiltration method utilized for transient transformation in *Nicotiana benthamiana*. The generation of simple insertion events for the T-DNA marked by the border (left and right) sequences in a binary vector is the hallmark of generating transgenics with a reduced frequency of transgene silencing [108]. Zheng et al. [109] have developed the conventional *Agrobacterium*-mediated transient gene expression system in which the seedlings and plantlets were co-cultivated with Murashige and Skoog medium containing *Agrobacterium* for several days exhibiting transient expression of the *β-glucuronidase* (*GUS*) gene. Further, it is a shorter and efficient way that can help reduce the time required for the development of rice transgenics [110].

Fischer et al. [111] have also reported that transient gene expression is a convenient method as compared to time-consuming stable transformation methods for the production of recombinant proteins. It has been reported that 100% transient transformation efficiency can be achieved in *Nicotiana benthamiana* and lettuce [112]. The biggest advantage of plant transient transformation is the gene expression for a short time, in which the target gene can express within 12 h and high expression could be attained within 2~4 days in the plant cells [113]. The non-integrated T-DNA copies remain transiently present in the nucleus that can transcribe and translate, leading to T-DNA transient gene expression. The transient genetic transformation can unveil the complex functional genomics in conifer species [114] and medicinal plants [115].

Li et al. [92] have developed a novel transient assay Fast Agro-mediated Seedling transformation (FAST) in the year 2009 which was based on co-cultivation of Arabidopsis seedlings with *Agrobacterium tumefaciens* in the presence of a Silwet L-77 surfactant. They have successfully expressed the constructs driven by different promoters in cotyledons but not roots of Arabidopsis seedlings in diverse genetic backgrounds through this approach. Another method AGROBEST (*Agrobacterium*-mediated Enhanced Seedling Transformation) has also been developed during the year 2014 by Wu et al. [94] to obtain high transient transformation efficiency in whole seedlings. Presently *Agrobacterium*-based transient transformation is a safe, high-level, and rapid transient transgene expression method [95,116,117,118]. Moreover, agroinfiltration is a promising technology for *in planta* transient expression of high-value recombinant proteins [119,120]. Also, *A. tumefaciens* based transient gene expression tends to affect plant signal transduction that could be used for studying various regulatory processes and defense mechanisms in the plant system [121].

## 7. Stable vs. Transient Genetic Transformation

The transient genetic transformation exhibits upto 21.8% high transformation efficiency as compared to 0.14% stable genetic transformation efficiency [99]. Stable transformation being labour intensive process that takes even 15 months [122] or even more for the production of transformed plants. On the other hand, transient transformation is a rapid method with a high expression level. The transient expression holds promise before implementing the tedious and time-consuming stable expression methods [30,94,123]. The transient transformation has several advantages over stable transformation as mentioned below: (i)Exemption of tissue culture [33](ii)Temporary expression [52](iii)T-DNA position effect elimination [33](iv)Simple, quick, economical, and effective [109](v)The transient transformation frequency was at least 1000-fold greater than that of stable transformation [29,95](vi)Transient transformation is often versatile, quick, and efficient as compared to stable transformation [99].

## 8. Agroinfiltration

The transient expression methods include transfection of protoplasts using PEG-mediated or electroporation [124], biolistics [125], and agroinfiltration [126], but agroinfiltration is a simple and effective method for transfer of the gene of interest in the host cell [118]. It involves direct delivery of recombinant agrobacteria carrying the gene of interest in the extracellular leaf spaces by physical or vacuum infiltration [25]. Agroinfiltration led to a transfer of single-stranded T-DNA from *Agrobacterium* to the plant cells which is trafficked through chaperones to the nucleus. During integration, a small percentage of T-DNA is integrated into the host chromosomes that lead to the development of stable transformation [36]. On the other hand, part of T-DNAs that do not integrate into chromosomes results in the high production of short-lived recombinant protein production [127]. The host factors (karyopherin α and VIP1), bacterial factors (VirD2, VirE2, and VirE3), and plant factor KU80 play a vital role in T-DNA integration in *Agrobacterium*-based genetic transformation methods [6]. Moreover, this method applies to leaf tissue as well as other plant organs, like fruits and berries and recalcitrant plant species [33,128]. Agroinfiltration has high transformation efficiency, high scalability, and ease of multiple transient expression assays with multiple transgenic vectors and multiple R genes carrying the gene of interest on a single leaf [129,130]. The transient expression of two resistance genes *Cf-9* and *Cf-4* has been observed in tomato plants through agroinfiltration approach in tobacco. It has also been observed that co-expression of *R* gene with the corresponding *Avr* gene trigger host-defence responses in a hypersensitive response in *Nicotiana tabacum*. It resulted in the formation of the necrotic sector through the expression of both genes in the overlapping region that led to the transient production of R and Avr proteins in the infiltrated leaf area through agroinfiltration [131]. Kapila et al. [7] have first worked on the intact petunia leaves for the expression of heterologous proteins through the infiltration of *A. tumefaciens* suspensions of harbouring a binary vector into leaf interstitial spaces. The gene transfer through agroinfiltration enabled the transfer of T-DNA in all cell layers of the leaf including intercellular spaces in the upper epidermis, lower epidermis, and palisade parenchyma, spongy parenchyma, and interstitial cell layers. Li et al. [92] have developed the transient agro-infiltration method for rapid production of recombinant protein, an attribute fundamental for functional genomics studies. The process of Agroinfiltration is temporary as it remains in the in vivo system for a limited period [132], therefore, there occurs no T-DNA integration into the plant genome [133,134]. The implementation of transient genetic transformation in the research studies will serve as a pilot experiment before following stable transformation which will provide information about the knock-out effect of the gene in question. It could be used in replacement of the in vitro transcription and cleavage assay of sgRNA and Cas9 protein of protoplast transfection method. It has been reported that *Agrobacterium*-based transient agroinfiltration can achieve success within 3 days without the need for any cumbersome procedures and equipment. Moreover, *Agrobacterium*-based transient approach agroinfiltration is the one that saves time, labour will lead to the temporary transformation which will make genome editing “safer technology”. A synonym term agroinoculation implies the use of a virus-based vector while agroinfiltration is based on the implementation of binary vector marks the difference between both approaches [135]. In comparison with the plant breeding based back crosses approach that has the potential to remove the CRISPR-Cas9 vector insertion and retain only the beneficial gene-edited alleles but agroinfiltration is an efficient approach that can be performed as a preliminary screening technique for somatic or germinal cells and later the stable transformation technique can be followed.

The transient protein production in the cytoplasm due to the absence of insertion of genes into the crop genome will have an exemption from transgenic regulation and hence greater public acceptance [131]. Agroinfiltration results in the direct creation of marker-free tobacco plants [136]. It is the method that results in the rapid functional analysis of transgenes directly [137]. The molecular and functional analysis can also be achieved through agroinfiltration [138]. The genetic background and the tissue culture conditions are the important measures for a successful agroinfiltration experiment [139].

## 9. Methods of Agroinfiltration

Agroinfiltration can be carried out through numerous methods like syringe infiltration, vacuum infiltration, hydrogen peroxide-based agroinfiltration, special agroinfiltration method, leaf disc vacuum infiltration, spray-based agroinfiltration, and detached leaf-based infiltration approach which will be briefly summarised here. Generally, agroinfiltration is carried out using two methods; syringe and vacuum infiltration applicable for a variety of plant species (Table 1, Table 2 and Table 3). Among these methods, vacuum infiltration requires specialized equipment but exhibits improved transformation potential or yield [98]. An advanced modified version of the vacuum infiltration technology namely leaf disc vacuum infiltration has also been developed that involves pulling off the syringe plunger for the creation of a small vacuum in the syringe [140]. Likewise, Xu et al. [141] have developed a special agroinfiltration method to attain gene expression within 5 days. This method involved the injection of the gene of interest into the interfaces between the adaxial epidermis and mesophyll using a plastic syringe with a needle. This leads to the formation of an agroinfiltration bubble (approximately 1 cm^2^ area) with a high ratio of agrobacteria carrying gene of interest to be infiltrated in the epidermal cells.

Shin and Park [142] have developed another agroinfiltration method namely hydrogen peroxide-based agroinfiltration. The use of this abrasive chemical induces wounds in the treated tissue which results in the improvement in the transformation efficiency in Chinese cabbage. A novel spray-based agroinfiltration is the technology that is applicable without the need of a vacuum chamber [143]. The detached leaf-based infiltration approach is also developed to overcome the necrosis, photo-bleaching, browning, and senescence cell death problem in intact leaves [144]. 

## 10. Factors Affecting Agroinfiltration

The success of agroinfiltration depends on numerous factors that need to be optimized to increase the transformation efficiency [97,140]. These factors influence the transfer of T-DNA from *Agrobacterium* to plant cells [145] and include plant genotype, type of explant, *Agrobacterium* strain, cell density in the inoculation medium, inoculation conditions, and co-culture. A comprehensive detail about these factors as discussed in previous reports has been presented in Table 1. Apart from these factors, numerous physical and molecular factors need to be considered during *Agrobacterium*-mediated transformation. The physical factors include ambient and leaf temperature, light source, pH, osmotic conditions, explant type, bacterial strain, density, and co-cultivation time that may affect the *Agrobacterium*-based transformation [146,147,148]. The infiltration time for 30 min has been observed to improve the transformation in Medicago leaves while less than 25 min was considered to be best for *Trifolium* leaves [149]. The surfactants such as Silwet L-77 or dark treatment after infiltration can improve the transformation efficiency during transient transformation [25]. The effects of the infiltration method, *Agrobacterium* strain, and age of the donor plant are the critical factors in the transient transformation studies [150]. Secondly, molecular factor involving foreign gene transfer followed by transgene expression may be altered by the inherent host plant mechanism of post-transcriptional gene silencing [151]. It has been observed that gene silencing results in the loss of protein expression so the co-expression of gene silencing inhibitor (e.g., p19 protein from tomato bushy stunt virus, NS1 protein from human influenza A virus and P1/HC-Pro protein of Tobacco etch potyvirus) can lead to 50 fold enhancement in the expression of the candidate protein through agroinfiltration process [152,153].

**Table 1 ijms-22-10882-t001:** Optimization of various parameters for agroinfiltration in plants.

Plant	Family	Cultivar/Genotype (s)	Target Gene	Tissue	Method	*Agrobacterium* Strain (s)	Binary Vector (s)	Optimization for Agroinfiltration	Detection Methods	Remarks	Reference
Model Plants											
*Arabidopsis thaliana* (Arabidopsis)	Brassicaceae	Columbia (Col-0) ecotype	*GUS*	Leaf	Syringe (needleless)	LBA4404, C58C1, GV3101, EHA105 and AGL-1	pCAMBIA1304	0.01% Triton X-100 or 0.01% Tween-20 LAB4404—Best *Agrobacterium* strain	GUS staining	Incubation of the infiltrated plants under short day conditions at high relative humidity maximize the gene expression	[91]
*Cucumis melo* L. (Melon)	Cucurbitaceae	-	*Nattokinase (NK)*	Fruit	Syringe (needle)	LBA4404	pPZP35S, pPZP35SN, pPZP35SNi, pPZPE8, pPZPE8N and pPZPE8Ni	Acetosyringone 0.2 mm and Codon-optimized synthetic NK gene	Quantitative Real Time PCR (qRT-PCR) analysis and fibrinolytic activity	High expression of recombinant NK gene	[154]
*Glycine max* (Soybean)	Fabaceae	Williams 82, Jack, JackX, ‘Peking’, L77-1863 and Williams	*GUS*	Leaf and seedlings	Syringe (needleless) and vacuum	A281 EHA105 LBA4404 Ach5 and J2	pCambia1305.1	Infiltration buffer (10 mM 2-(N-morpholino) ethanesulfonic acid sodium salt, 10 mM MgCl_2_, 100 µM acetosyringone) with dithiothreitol and 30 s sonication	GUS assay	Increase in the agroinfiltration-mediated GUS expression	[155]
*Nicotiana benthamiana* (Tobacco)	Solanaceae	-	*GUS*	Leaf	Syringe (needleless)	EHA105	pCAMBIA1301	20 µM azacytidine, 0.56 mM ascorbate and 0.03% (*v*/*v*) Tween-20	qRT-PCR	At about 6-fold higher transient gene expression	[98]
		-	*GUS*	Leaf	Syringe	AGL1, C58C1 and LBA4404	pEAQ-GSN	Acetosyringone (500 μM), Lipoic acid (5 μM), Pluronic F-68 (0.002%) and 37 °C heat shock	GUS assay, Enzyme-linked immunoassay (ELISA) and Polyacrylamide gel electrophoresis (PAGE) analysis	Around 3.5-fold higher levels of absolute GUS protein compared to the pEAQ-HT deconstructed virus vector platform	[118]
		Wild-type	*Green fluorescent protein* (*GFP*)	Leaf	Syringe (needleless)	EHA105, LBA4404, AGL0 and AGL1	pCAMBIA(gfp)1302	Best *Agrobacterium* strain-AGL0 and EHA105, acetosyringone 450–600 μM, viral protein HC-Pro, Leaf ageing	GFP imaging	High gene expression was observed in the youngest leaf	[156]
		Wild type non-transgenic plants	*Anthrax receptor decoy protein (immunoadhesin)* and *CMG2-Fc*	Whole plants and detached Leaf	Vacuum	-	pBIN and pCB302	Number of viral suppressors of post-transcriptional gene silencing constructs: p1, p10, p19, p21, p24, p25, p38, 2b, and HCPro	ELISA, Bradford assay and Western Blotting	p1 exhibit maximum gene expression contributing towards post transcriptional gene silencing	[157]
		-	*Ave1* and *Ve1*	Leaf	Syringe	GV3101	Gateway-compatible binary vectors	Gateway-compatible binary vectors improve agroinfiltration efficiency	Polymerase Chain Reaction (PCR)	*Ve1*-mediated resistance against *verticillium*	[158]
		-	*Firefly luciferase*	Leaf	Syringe (needleless)	C58C1 (pGV2260)	pExp35S-LUC	Hierarchical design of promoter, leaf, plant and sampling position	Luciferase activity	Best result through sampling more positions on the same leaf	[159]
		Transgenic plants	*AC1*, *AC2*, *AC4* from *DNA-A* and *BC1* from *DNA-B* of African cassava mosaic virus (ACMV)	Leaf	Syringe (needleless)	GV3101	RNA interference constructs	ACMV-Cameroon:DO2:1998 transient protection assay	Electrophoresis, southern and northern hybridizations	Systemic movement of the silencing signal	[160]
		-	*Hemagglutinin ectodomain* *derived from influenza A virus strain A*	Detached leaf	Vacuum	GV3101	pMP90	Variation in the duration of water removal treatment from 0.7 to 4.4 h	Sodium dodecyl sulfate polyacrylamide gel electrophoresis (SDS-PAGE)	Improvement in recombinant hemagglutinin yield	[161]
		-	*Human epidermal growth factor*	Leaf	Syringe (needleless)	GV3101	pBYR2e-hEGF	Expression vector carrying different hEGF constructs, *Agrobacterium* cell density (0.2, 0.4, 0.6, and 0.8) at OD600	ELISA	Production of recombinant hEGF protein	[162]
		-	*GFP*, *DsRed fluorescent protein*, *Yellow fluorescent protein* (*YFP*) and *Cyan Fluorescent Protein* (*CFP*)	Leaf	Syringe (needleless)	EHA105	pBYKEAM or pBYKEAM2	Plant expression vectors	SDS-PAGE, Fluorescence Imaging, Western Blotting, ELISA	High level production of monoclonal antibodies	[163]
*Nicotiana tabacum* (Tobacco)	Solanaceae	*N. tabacum* cv. Samsun and Xanthi, and *N. benthamiana*	Human interferon-*γ (hIFN-γ)* protein	Leaf	Syringe (needleless)	EHA101, GV3101, and LBA4404	pGEM-hIFN-γ	Best *Agrobacterium* strain GV3101 with OD600 of 1.0 and acetosyringone 200 µM at 4 days post agroinfiltration	Reverse Transcription polymerase chain reaction (RT-PCR), qRT-PCR, SDS-PAGE, Western Blotting, ELISA	Bioactive hIFN-γ protein production	[164]
*Pisum sativum* (Pea)	Fabaceae	*Pisum sativum* and *Medicago sativa* plants	*Salivary gene*	Leaf	Syringe and vacuum	AGL-1	pEAQ-HT-DEST1	Screening of a range of pea cultivars	Protein extraction and Western-Blotting	Increase aphids fecundity	[165]
*Solanum lycopersicum* (Tomato) and *Nicotiana benthamiana* (Tobacco)	Solanaceae	MicroTom, a dwarf tomato cultivar	*GFP*	Leaf	Syringe (needleless)	EHA105	pCASGFPt (control GFP), pOsAPP1GFP (pGFPTag16) and pOs*ZF*1GFP (pZF1gfp)	Testing of agroinfiltration by expressing GFP fusions of the putative antiphagocytic protein 1 (*APP1*) (Os*APP*, LOC_Os03g56930) and ZOS3-18—C2H2 zinc-finger protein (Os*ZF1*, LOC_Os03g55540)	GUS staining	Subcellular localization of proteins	[166]
Floricultural crops											
*Cannabis sativa* L. (Hemp)	Cannabaceae	Fedora 17, Felina 32, Ferimon, Futura 75, Santhica 27 and USO31	*Phytoene Desaturase* (*PDS*) and *GUS*	Plant tissue-mature leaf discs, mature leaf, pollen sacs, anthers, sepals, pollen sac clusters, filaments, pollen grains, nonglandular trichomes, female flowers and pistil	Vacuum	EHA105, LBA4404 and GV3101	pEarleyGate 101-uidA	Silwett L-770 (0.015%), ascorbic acid (5 mm) and sonication of 30 s followed by a 10-min vacuum treatment	qRT-PCR	Highest *GUS* expression in the leaf, stem, root tissues, male and female flowers	[167]
*Eustoma russullianus* (Lisianthus)	Gentianaceae	-	*GUS*	Pollen	Vacuum	LBA4404	pBI121	Sucrose 7–15% pH 5.5–7.0 Temperature 20–27 °C	GUS assay, Southern hybridization and RT-PCR	Pollen transformation	[168]
*Gerbera jemosonii* (Gerbera)	Asteraceae	Express and White Grizzly	*GUS*, *GFP*, *iris-dfr and petunia-f3′5′h*	Flower	Syringe and vacuum	GV3101	pCambia/*dfr* and pFGC5941	Vacuum infiltration prove to be the best method	GFP and GUS assay	Change in the anthocyanin pigment	[169]
*Piper colubrinum Link* (Black pepper)	Piperaceae	-	*GUS* and *Serine threonine protein kinase (STPK) gene*	Detached Leaf	Vacuum	EHA 105	pCAMBIA 1305.2 and pHELLSGATE	Higher vacuum up to 400–600 mm Hg increased infiltration transformation efficiency	qRT-PCR	Silencing of STPK gene	[170]
*Vitis vinifera* L. (Grapevine)	Vitaceae	Sugraone, Aleatico, Moscato Giallo and Aglianico	*Free GFP and (mRFP1), GFP::HDEL, GAPA1::YFP and b::GFP*	Leaf	Syringe (needleless)	LBA4404, GV3101 and AGL1	pBI121, pBIN-m-gfp5-ER, pAVA554, pRSET-mRFP1, pAVA554 and pGreen 0029	Combination of sugraone cultivar and the GV3101 showed high gene expression	GFP imaging	Compatibility between *Agrobacterium* strain and genotype exhibited high transient gene expression	[139]
*Vitis vinifera* (Grapevine)	Vitaceae	Cabernet Sauvignon, Cinsault, Muscat Ottonel and Syrah	*GUS*, *GFP* and *stilbene synthase*	Leaf	Syringe (needleless) and vacuum	C58C1	pBIN19 and pBINY53	Presence of additional virulence factors like virG and virE promote infiltration	RT-PCR, GUS staining and Fluorescence microscopy	Vacuum infiltration better than syringe infiltration	[171]
Horticultural crops											
*Maesa lanceolata* (False assegai)	Primulaceae	-	*GFP*	Leaf	Syringe	C58, EHA101, EHA105, LBA4404, GV3301, GV2260 and pMP90	pK7FWGF2	*A. tumefaciens* strain LBA4404 at an OD600 = 1.0 in the presence of 100 µM acetosyringone and in the absence of viral suppressor construct	PCR	Saponin production	[172]
*Malus domestica* Borkh-(Apple), *Pyrus communis* L. (Pear)	Rosaceae	Apple ‘Gala’ and Pear ‘Conference’	*GUS*	Leaf	Vacuum	EHA105	pBBR1MCS-5	Silwet L-77 at a low concentration (0.002% *v*/*v*)	Optimising through 10 different binary plasmids and *A. tumefaciens* inoculations	Transformation efficiency between 50 and 80%	[173]
Leguminous crop											
*Mucuna bracteata*	Fabaceae	-	*Anti-toxoplasma immunoglobulin*	Leaf	Vacuum	GV3101	pTRAkcHcLcTg130	High expression in bottom trifoliate leaf at 2 days post-infiltration	Western blotting and ELISA	Transient expression in *M. bracteata*, was two-fold higher than the model *Nicotiana benthamiana* plant	[174]
Vegetable Crop											
*Spinacia oleracea* (Spinach)	Chenopodiaceae	Korean cultivar Sakyechul	*GUS*	Leaf	Syringe (needleless) and vacuum	EHA105, LBA4404 and GV2260	pB7WG2D-GUS	*Agrobacterium* GV2260 strain suspension at OD600 of 1.0	qRT-PCR	Increased efficiency, duration of gene expression and protein accumulation	[97]
Vegetable and model crops											
*Lactuca serriola* and *L. sativa* (Lettuce), *Lycopersicon esculentum* (Tomato), *N. benthamiana* (Tobacco) and *Arabidopsis thaliana* (Arabidopsis)	Asteraceae, Solanaceae and Brassicaceae	Wild lettuce LS102, cultivated lettuce cv. Valmaine and cv. Mariska, tomato-Rio Grande 76R, Arabidopsis-Columbia-0	*GUS*	Leaf	Syringe (needleless)	42 wild strains	pCB301‘empty’, tobacco etch virus-P1/HcPro, turnip mosaic virus-P1/HcPro and P19 from tomato bushy stunt virus	Best *Agrobacterium* strain C58C1	GUS assay	High gene expression in lettuce as compared to *Nicotiana benthamiana*	[88]
*Nicotiana tobacum* (Tobacco), *Solanum tuberosum* (Potato) and *Lactuca sativa* (Lettuce)	Solanaceae and Asteraceae	*Nicotiana tobacum* cv. Xanthi, *Solanum tuberosum* cv. Agria	*Human growth hormone*	Leaf	Vacuum	pGV3850	pBin19	Time span of infiltration upto 35 min	Western blotting and ELISA	High production of recombinant hGH protein in tobacco and potato as compared to lettuce	[175]

Cytoplasmic RNA silencing may occur during agroinfiltration transient expression [176]. Recently, Norkunas et al. [118] have identified the effects of chemical additives and heat shock pretreatment to confer improved transformation through agroinfiltration in *Nicotiana tabacum*. The construct comprised of the geminiviral replication system and a double terminator including heat shock protein terminator combined with an extensin terminator can also increase transient protein expression during the agroinfiltration process [177]. Hyperosmotic pretreatment with sucrose (25% *w*/*v*) containing 1/2 MS solution (pH 5.8) for 3 h before agroinfiltration greatly improved transient expression efficiency [109]. Sheludko [29] reviewed the optimization of transient expression protocol for high-scale protein production. The leaf position of apical leaves showed high transient *GFP* expression [140]. The transient transformation efficiency of 99–100% in recalcitrant banana plants was achieved by using hydroponic solution followed by syringe-based infiltration [178]. 

## 11. Advantages of Agroinfiltration Method over Other Transient Genetic Transformation Methods

Various genetic transformation methods for transient genetic transformation in plants include PEG-mediated, biolistics, viral vectors, and agroinfiltration. Agroinfiltration exhibits numerous advantages over the above-mentioned transient methods which are briefly summarized here. Agroinfiltration process does not involve isolation of protoplast which is a pre-requisite for the PEG-mediated genetic transformation. Further, agroinfiltration is a simple and cost-effective method that simply involves the infiltration of transgenes into the intercellular layers of plant cells [7,118]. Moreover, agroinfiltration is less time-consuming compared to the protoplasts-based genetic transformation [179]. 

The high-velocity biolistics method also exhibits a transient expression of an introduced gene(s) in the plant cells. This method was first demonstrated in the laboratory of J.C. Sanford at Cornell University in the late 1980s [8]. But this rapid and versatile method suffers from a major limitation of the requirement of special equipment PDS-1000/He [32]. Moreover, the transformation efficiency of biolistics is relatively low as compared to agroinfiltration [180]. The efficiency of promoters can be tested efficiently in agroinfiltration as compared to the biolistics genetic transformation method [181]. 

The transient expression was also achieved using viral vectors but this method suffered from biosafety and construct-size limitation issues for the insertion of transgenes [152,182]. Contrastingly, agroinfiltration involved the infiltration of the whole leaf with one or multiple target DNA constructs or introducing multiple constructs into the different regions of the same leaf [135]. Thus, agroinfiltration is a better suited transient genetic transformation method compared to biolistics, viral vectors, and PEG-mediated methods in plants. 

## 12. CRISPR-Cas9 Based Genome Editing via Agroinfiltration

The CRISPR/Cas9 system can be effectively coupled with the agroinfiltration transient expression system thereby employing the delivery or use of Cas9 and sgRNA for targeted site modifications in plants (Figure 2). The conjugation of CRISPR/Cas9 (primary choice for plant genome editing) and *Agrobacterium*-mediated transformation (most preferred method for gene delivery) has the power to deliver the CRISPR/Cas9 DNA constructs efficiently to plants without integration into the plant cells [183]. Generally, agroinfiltration forms a part of the initial screening used as a preliminary experiment to identify whether a particular plant tissue type is amenable to transformation or not which is indirectly validated through the sgRNA expression (genome editing) or genetic transformation (*gfp* gene or other gene studies). The studies linked to the use of agroinfiltration-based CRISPR-Cas9 genome editing in plants have been given in Table 2. The non-transmissible agroinfiltration approach serves as the first line of experiments to test the extent of transient genome editing before initiating the elaborate genome editing experiments. Contrastingly, a recent comprehensive review is available that emphasizes the use of *Agrobacterium*-based floral dip stable transformation for genome editing [4]. The focus on the *Agrobacterium*-based transient gene transformation agroinfiltration method in genome editing bears better advantages over the *Agrobacterium*-based stable genetic transformation methods. The transient agroinfiltration approach is widely employed for the prediction of gRNA efficiency in genome editing constructs in vivo (Table 2). The combinatorial approach of CRISPR-Cas editing and agroinfiltration showed the result in the transformed tissue within 3 days after infiltration that serve as a reliable assay for testing sgRNAs under native conditions [184].

**Table 2 ijms-22-10882-t002:** Agroinfiltration based genome editing strategies for plants.

Plant	Family	Cultivar/Genotype	Target	Tissue	Stage	Method	*Agrobacterium* Strain	Promoter	Cas9-Codon Optimised	Detection Assay	Mutation Rate	Reference
Model plants												
*Arabidopsis thaliana* (Arabidopsis)	Brassicaceae	Columbia-0	*PDS*	Seedlings	2-week-old	Syringe (needleless)	GV3101	CaMV35SPDK and AtU6	Yes	PCR and Sanger sequencing	2.7%	[185]
*Arabidopsis thaliana* (Arabidopsis)	Brassicaceae	Transgenic plants	*GFP*	Leaf	4-week-old	Syringe (needleless)	C58 and EHA105	CaMV35S and AtU6	Yes	Fluorescence confocal microscopy and Sangersequencing	-	[186]
*Nicotiana benthamiana* (Tobacco)	Solanaceae	-	*PDS*	Leaf	5-week-old	Syringe (needleless)	GV3101	CaMV35SPDK and AtU6	Yes	PCR and Sanger sequencing	4.8%	[185]
		-	*PDS*	Leaf	3–4 weeks	Syringe (needleless)	AGL1	CaMV35S and AtU6	Yes	PCR and restriction enzyme assay	2.1%	[187]
		-	*PDS*	Leaf	3–4 weeks	Syringe	GV3101	CaMVE35S	-	RT-PCR and Sanger sequencing	12.7–13.8%	[188]
		-	*PDS*	Leaf	3–4 weeks	Syringe	GV3101	BS3 and uid	Yes	qRT-PCR and Sanger sequencing	-	[189]
		Transgenic KQ334 plant	*NbPDS3 and isopentenyl/dimethylallyl diphosphate synthase (NbIspH) genes*	Leaf	Six-leaf stage	Syringe	GV3101	CaMV35S and U6	Yes	RT-PCR and Sanger sequencing	*PDS*-85%, *IspH*-75%	[190]
		-	*PDS*	Leaf	3-week-old	Syringe (needleless)	AGL1	J23119	-	PCR, Sanger and Illumina sequencing	3–18%	[191]
		Transgenic plants	Six sites Bean yellow dwarf virus genome—*Rep binding site (RBS), hairpin, nonanucleotide sequence and three Rep motifs*	Leaf tips	5-week-old	Syringe	GV3101	Double 35S promoter and AtU6 or At7SL RNA polymerase III promoter	-	qRT-PCR and Illumina sequencing	0.03–70.01%	[192]
		-	*Xylosyltransferase gene*	Leaf	5–6 weeks	Syringe (needleless)	GV3101	Nopaline synthase	Yes	PCR, restriction digestion analysis and Sanger sequencing	12.1%—*XT1* and 9.9%—*XT2*	[193]
*Nicotiana tabacum* (Tobacco)	Solanaceae	Cas9-overexpressing transgenic lines	*PDS and proliferating cell nuclear antigen gene (PCNA) genes*	Leaf	-	Syringe	GV3101	CaMV 35S	Yes	T7 endonuclease 1 based assay, restriction digestion analysis and Sanger sequencing	-	[194]
		-	*PDS*	Leaf	-	Syringe (needleless)	EHA105 and GV3101	AtU6	Yes	PCR and Sanger sequencing	-	[195]
		-	*PDS*	Leaf	3–4 weeks	Syringe	GV3101	35SPDK	Yes	Flourescent microscopy and PCR analysis	-	[196]
		Wild-type	*GFP*	Leaf	4-week-old	Syringe (needleless)	C58 and EHA105	CaMV35S and AtU6	Yes	Fluorescence confocal microscopy and Sanger sequencing	-	[186]
*Solanum lycopersicum* (Tomato)	Solanaceae	Transgenic plants	*Immunity associated genes*	Leaf	4-week-old	Syringe (needleless)	1D1249	U6 promoter	-	PCR and Sanger sequencing	61.5%	[96]
*Cereal crop*												
*Sorghum bicolor* (Sorghum)	Poaceae	Tx430 plants	*GFP*	Leaf	3–4 weeks	Syringe (needleless)	GV3101	CaMV 35S and maize Ubiquitin 1	Yes	Fluorescence microscopy	-	[184]
Horticultural crops												
*Citrus sinensis* (Sweet Orange)	Rutaceae	Valencia cultivar	*PDS*	Leaf	Three-year-old	Syringe	-	CaMV 35S	-	PCR and Sanger sequencing	3.2–3.9%	[197]
*Citrus paradisi* (Grapefruit)	Rutaceae	Wild type Duncan and transgenic plants	*Canker susceptibility gene* (*CsLOB1*)	Leaf	-	Syringe (needleless)	EHA105	Cassava vein mosaic virus promoter and CaMV 35S promoter	-	Illumina sequencing	3.58–88.78%	[198]
*Fragaria × ananassa* (Strawberry)	Rosaceae	*Fragaria vesca* (cv. Reine des Vallées) and *F. × ananassa* Duch. (cv. Camarosa)	*Tomato MADS box gene6*	Fruit	Green stage	-	AGL-0	35SCaMV and AtU6-26	-	PCR and Sanger sequencing	-	[199]
*Floricultural crop*												
*Papaver somniferum* L. (Opium poppy)	Papaveraceae	*P. somniferum* (cv. Ofis-95)	*3′-hydroxyl-N-methylcoclaurine 4′-O-methyltransferase (4′OMT2)*	Leaf	-	Syringe (needleless)	EHA105 and GV3101	AtU6	Yes	PCR and Sanger sequencing	-	[195]
*Leguminous crop*												
*Vigna unguiculata* [L.] Walp. (Cowpea)	Fabaceae	Transgenic plants	*Meiosis genes* i.e., *SPO11-1*, *REC8*, and *OSD1*	Detached leaflets	3–4 weeks	Syringe (with and without needle)	AGL-1	Arabidopsis ubiquitin 3, RPS5a and AtU6-26	-	PCR and Illumina sequencing	1%	[144]
*Vegetable crops*												
*Brassica oleracea* var. *capitata* f. *Rubra* (Purple cabbage)	Brassicaceae	Rebecca F_1_ and Huzaro F_1_	*Centromere-specific* *histone H3*	Leaf	6-week-old	Syringe	GV3101	CaMV35S and AtU6	Yes	PCR and Illumina HiSeq sequencing	0.07–14.42%	[200]
*Dioscorea alata* (Yam)	Dioscoreaceae	Transgenic plants	*PDS*	Leaf	2 months old	Syringe	EHA105 and LBA4404	DaU6 promoter, maize ubiquiti and CaMV35S	-	PCR and Sanger sequencing	83.3%	[201]

Among prevailing genome editing tools *viz*. CRISPR-Cas widely acceptable tool other than ZFNs and TALENs for crop improvement [131,202,203]. Most of the researchers targeted the *PDS* gene required for carotenoid biosynthesis. Li et al. [185] have utilized agroinfiltration for targeted genome modifications in the *AtPDS3* and *NbPDS* genes of *Arabidopsis thaliana* and *Nicotiana benthamiana*. The *PDS* disruption enhanced the chlorophyll oxidation that led to the development of a visible photobleached phenotype [104,185]. The agroinfiltration was used for CRISPR/Cas9-mediated genome editing targeting *PDS* gene in various plant species [179,185,187,188,189,190,191,194,195,196,197,201]. Syombua et al. [201] have employed the *Agrobacterium*-based transient transformation method; agroinfiltration and genome editing for Yam (*Dioscoreaalata*) plant genetic improvement. The infiltration of pCas9-gRNA-PDS in leaves tends to show bleached patches that can be analyzed through microscopic examination of an infiltrated leaf section to confirm the transient knockout of the *PDS* gene, and the effectiveness of both agroinfiltration and genome editing approaches.

Along with *PDS*, researchers have also utilized another visible marker GFP to analyse the success of the agroinfiltration-based genome editing [186]. The use of non-functional GFP containing the target site cleaved by a Cas9/sgRNA complex resulted in the creation of functional *GFP* genes to report the success of agroinfiltration-based genome editing [186]. The conversion of frame-shifted GFP to visible GFP due to CRISPR-Cas9 genome editing marks the achievement in agroinfiltration-based genome editing that can be analyzed through fluorescence microscopy of the infiltrated section [201]. Later on, the transient expression can be followed by stable genetic transformation that tend to show phenotype such as dwarf phenotype, albino shoot with a bushy phenotype, and variegated albino plantlets [201]. Besides, fluorescent markers, vectors namely pGD vectors carrying GFP and DsRed elements also ease the interpretation of co-localization and protein-protein interactions in dicots in the agroinfiltration in plants [204]. Moreover, pBYR2HS vector in agroinfiltration applies to leaves and fruits [82]. Sharma et al. [184] have developed a simplified transient transformation assay using agroinfiltration in sorghum and confirmed the gene editing in sorghum leaves using GFP as a marker. The agroinfiltration-based genome editing was used to monocot sorghum which is an unlikely host for *Agrobacterium* due to the occurrence of epidermal cuticular wax, high silica content, and low volume of intercellular space. However, it was observed to be a successful technique for achieving genome editing in the sorghum. The sgRNA efficiency in orange and purple cabbage has been tested using agroinfiltration [197,200]. Likewise, agroinfiltration transient approach has been tested followed by implementation through stable transformation for gene characterization studies in strawberry [199] (Knockout of *Tomato MADS box gene6* generated mutant lines that deciphered its role in anther development), yam [201] (Knockout of *PDS* gene resulted in the development of albino plants) and cowpea [144] (Knockout of meiosis genes *viz. SPO11-1*, *REC8*, and *OSD1* showed defects in male and female meiocytes displaying its role for the induction of mitosis from meiosis). Most of the researchers reported the leaf (Table 1) as a host for agroinfiltration-based genome editing while Baltes et al. [192] reported the agroinfiltration-based genome editing using leaf tips with 0.03–70.01% mutation rate conferring geminiviruses resistance.

Further, the repercussions of the genome edits under in vivo conditions are always subtle and not easily detectable. It has been evident that the predicted sgRNAs often have different editing efficiencies in homozygous and heterozygous conditions in each cell. Also, the variation in the knock-outs or sequences in any of the genes after genome editing events will be more useful to study the impact of genome editing or its efficiency rather than obtaining variations in the phenotypic characters. Moreover, the variation in the sequence can be easily identified through Sanger’s sequencing technique under in vitro conditions. Moreover, in silico prediction tools have been utilized to analyze the editing efficiency based on the choice and sequence feature of the sgRNA [184].

## 13. Modifications of Agroinfiltration-Based Genome Editing

Several reports showcasing the use of modified methods for agroinfiltration-mediated genome editing in plants have appeared over the recent decade. Jia and Wang [197,205] have published the first report on targeted genome modification of citrus through *Xanthomonas citri* subsp. *citri* (*Xcc*)-facilitated agroinfiltration to enhance the transient protein expression in a recalcitrant citrus species- Valencia sweet orange. They have recorded the involvement of PthA4Type III effector molecule and transcriptional activator-like effector leading to cell division (hyperplasia) and enlargement (hypertrophy) to be responsible for improved efficacy and success of *Xcc*-facilitated agroinfiltration. Jia and Wang [197,205] employed *Xcc*-facilitated agroinfiltration to deliver Cas9, along with a synthetic sgRNA targeting the phytoene desaturase (*CsPDS*) gene, into sweet orange and confirmed the mutation at the targeted location through DNA sequencing. The CRISPR-Cas9 technology employed agroinfiltration to confer resistance to tobacco rattle virus, bean yellow dwarf virus, and citrus canker accompanying targeting of numerous endogenous plant genes (Table 1) [144,190,192,193,194,195,198,199,200,206].

Several researchers have employed leaf for both syringe and vacuum infiltration. However, Juranic et al. [144] have illustrated the use of detached leaf rather than the intact leaf. They have interpreted that agro-infiltration of the intact leaves results in a variety of undesirable symptoms including necrosis, photo-bleaching, browning, and senescence or cell death. Hence, they have performed a detached leaf assay of leaflets with fluorescent expression constructs with the objective of meiosis-knock out for asexual seed induction in cowpea to identify gene-specific mutations.

Along with agroinfiltration, researchers have utilized the co-cultivation method as a transient transformation method [207,208]. Li et al. [207] have used this method and successfully employed the CRISPR/Cas9 technology to knock out the self-incompatibility-related gene *SRK* in Chinese cabbage with 10.83% transformation efficiency.

## 14. Other Applications in Plants

Agroinfiltration has enormous applications in plants (Table 3). It has been utilized for transient expression of transcription factor [209], antibody, and recombinant protein production [210,211]. Agroinfiltration is also applicable to effectoromics field involving rapid resistance and a virulence gene discovery [212]. Agroinfiltration has played a major role in a variety of studies including the gene silencing [122,213,214,215,216,217], elicitor identification [218,219], resistance mechanism [220,221,222,223], promoter characterization [224,225], expression studies [226], identification of plant receptor(s) [227], antigen-antibody interactions [228,229,230,231], vaccine production, protein ubiquitination [232], protein degradation [233], pigment [209] and phyto-sensing studies [234]. It has the potential to give a scrutinizing view of the plant system efficiently in the present times.

**Table 3 ijms-22-10882-t003:** Application of agroinfiltration in plants other than genome editing after the discovery of transient expression by Kapila et al. [7].

Plant	Target	Family	Material	Promoter	*Agrobacterium* Strain	Method	Outcome	Reference
Model plants								
*Medicago truncatula* (Barrel clover)	*LEGUME ANTHOCYANIN PRODUCTION 1 (LAP1) transcription factor*	Fabaceae	*Medicago truncatula* cv. R108	CaMV35S	GV3101	Syringe	Accumulation of anthocyanin pigment	[209]
*Nicotiana benthamiana* (Tobacco)	*C5-1 murine antibody*	Solanaceae	Seeds obtained from National tobacco germplasm	CaMV 35S	AGL1	Syringe	Recombinant protein production at lab scale	[211]
	*GFP transgene*	Solanaceae	-	CaMV35S	-	Syringe	Systemic silencing of a *GFP* transgene	[214]
	*epiGFP* (without GFP integration)	Solanaceae	Stably integrated GFP transgene (intGFP) transgenic plants	CaMV35S	-	Syringe	Systemic silencing through interaction between *epiGFP* and *intGFP*	[215]
	*Ubiquitin ligase-associated protein SGT1*	Solanaceae	Transgenic plants	CaMV35S	-	Syringe	Proof of *SGT1* that is required for host and nonhost disease resistance in plants	[222]
	*mGFP5-er, Bt Cry1Ac, and* *BoPI transgene genes*	Solanaceae	-	CaMV 35S	GV3850	Syringe	Ease the detection of candidate insect resistance transgenes	[223]
	*Chimaeric human β1,4-galactosyltransferase*	Solanaceae	Wild-type	CaMV 35S, Rubisco, plastocyanin	R612, R610, R621, R622 and 35SHcPro	Vacuum	High-yield antibodies production with human-like N-glycans	[230]
	*Haemagglutinin gene*	Solanaceae	-	Plastocyanin, chimeric double 35S	AGL1	Vacuum	Production of an influenza vaccine	[235]
	*E3 ligase Constitutive photomorphogenic1 (COP1) and its substrate HY5*	Solanaceae	Wild-type	-	EHA105 and ABI	Syringe	Detection of protein ubiquitination	[232]
	*Foot-and-mouth disease (FMD) virus P1-polyprotein (P1) and VP1 (viral capsid protein 1) and E. coli glutathione reductase (GOR)*	Solanaceae	-	CaMV35S	LBA4404	Syringe	Recombinant VP1 protein degradation	[233]
*Nicotiana* species (Tobacco)	*Cauliflower mosaic virus Gene VI*	Solanaceae	*N. edwardsonii* and *N. clevelandii*	CaMV35S	C58	Syringe (needleless)	Identification of gene VI protein elicitor	[219]
*Nicotiana sylvestris* (Tobacco)	*Class I chitinase A gene CHN48 transgene*	Solanaceae	Wild type and transgenic plants	CaMV35S	-	Syringe	Transgene silencing	[213]
*Nicotiana tabacum* (Tobacco)	*N gene*	Solanaceae	Samsun NN and nn plants	-	-	Syringe	Identification of the Tobacco Mosaic Virus elicitor	[218]
	*Avr9 and Avr4-Tobacco* *Cf-9 and Cf-4-tomato*	Solanaceae	*N. tabacum* cv. Petite Havana and transgenic tobacco lines	CaMV35S	MOG101	Syringe (needleless)	Co-expression of the *Avr4/Cf-4* gene pair confer resistance	[220]
	*Rx2, AC15*	Solanaceae	Tetraploid potato cultivars BZURA (*Rx2* genotype) and three susceptible potato accessions (*rx* genotype)	CaMV35S and Rx1	C58C1		Isolation of *Rx* resistance genes	[221]
	*Stress-responsive as-1 and heat shock elements, yeast GAL4 transactivation system, two promoters of pathogenesis-related genes as well as a heat shock promoter*	Solanaceae	*Nicotiana tabacum* var. Xanthi nc		EHA 105	Syringe	Identification of the *cis*-regulatory regions in promoters	[224]
	*Human lactoferrin*	Solanaceae	-	MPr1163 and CaMV E-35S	LBA4404	Syringe	Efficient use of chimeric promoter MPr1163 for the expression of heterologous protein	[225]
	*Collagen and chimeric P4H genes*	Solanaceae	-	L3, 1287	A1286 and A1284	Vacuum	Improvement in expression of collagen	[226]
	*Pseudomonas syringae pv. phaseolicola harpin (HrpZPsph) gene*	Solanaceae	*N. tabacum* cv. W38 TetR, *N. tabacum* cv. Xanthi, and *N. benthamiana*	CaMV35S	C58C1	-	Detection of plant cellular receptor(s) for harpin is extracellular	[227]
	*T84.66/GS8 diabody*	Solanaceae	*N. tabacum* cv. Petit Havana SR1	CaMV35S	GV3101	Vacuum	Production of A carcinoembryonic antigen-specific diabody	[228]
	*Human chorionic gonadotropin* (*hCG*)	Solanaceae	*Nicotiana tabacum* cv. Petite Havana SR1	-	GV3101	Vacuum	Production of recombinant antibodies against *hCG*	[229]
	*Foot and Mouth Disease Virus (FMDV) Coat Protein*	Solanaceae	-	CaMV35S	GV3101	Syringe	Production of recombinant antigen of FMD	[231]
*Glycine max* (Soybean)	*Coatomer subunit alpha (COPA) and aquaporin 9 (AQ9) genes*	Fabaceae	-	CaMV35S	EHA 105	Mechanical abrasion using carborundum	RNA interference against *Tetranychus urticae*	[217]
*Floricultural crops*								
*Vitis vinifera* L. (Grapevine)	*Grapevine gene VvPGIP1*	Vitaceae	Cabernet franc	CaMV35S	GV3101	Vacuum	Transient gene silencing	[216]
*Antirrhinum majus* (Snapdragon)	*AS1 and 4′CGT genes*	Plantaginaceae	-	-	LBA4404	Syringe	Conversion from white to pale yellow petals	[236]
*Horticultural crop*								
*Fragaria × ananassa* (Strawberry)	*Chalcone synthase gene*	Rosaceae	*F.* × *ananassa* cv. Elsanta	CaMV35 S	AGL0	Syringe	Gene silencing	[122]
Vegetable crops								
*Raphanus sativus* L. (Radish)	*Staphylococcal enterotoxin B (SEB) genes*	Brassicaceae		CaMV35S	LBA4404	Syringe	Production of Leaf-Encapsulated Vaccines	[237]
*Solanum melongena* L. (Eggplant)	*Hydroxycinnamoyl CoA-quinate transferase gene*	Solanaceae		p19 protein of Tomato bushy stunt virus (native promoter)	GV3101	Syringe (needle)	Improvement in chlorogenic content	[238]

## 15. Limitations

Although a useful technology, there are few pitfalls of Agroinfiltration which needs to be overcome through optimization of the genetic transformation factors while conducting the study as mentioned in Table 1 [239]. The major limitation of the transient expression is that it cannot be used for large-scale commercial production in seed banks due to its transient expression [240]. The complex configuration and organization of various types of plant cells foist varying low efficiency to the agroinfiltration process [33]. Also, high variability in the transformation efficiency depends on the type of *Agrobacterium* strains, target plant species, and specific tissues. Among the most popular methods of *Agrobacterium*-based genetic transformation, the syringe infiltration technique suffers from poor scalability that hinders experimental success [32]. Further, the plant defence responses are also a critical factor in genetic transformation studies [52]. Molecularly, the post-transcriptional gene silencing limits transient expression. Therefore, several viral suppressors like p19 protein of tomato bushy stunt virus have been introduced to enhance the transient expression [152].

## 16. Bio-Safety and Commercialization Aspects

The critical concern in plant biotechnology is the acceptance of genetically modified or genome-edited plants by consumers. Hence, biosafety and regulatory concerns need to be addressed before plant-based research studies. The generation of very high expression levels in a contained facility in an agroinfiltration-based transient approach minimizes the chances of biosafety-related risk [123]. Moreover, this technique has simple operation steps [241]. Although USA has already approved gene editing through *Agrobacterium*-mediated transformations yet the advent of targeting the somatic cells i.e leaves does not allow the foreign gene to be transferred in the next generation, hence, no chance of transgene to be transferred towards the next generation. The biosafety limitations of the genetically modified organisms (GMOs) generated through genetic engineering technology [242] can be efficiently dealt via agroinfiltration transient approach due to its ability to retain transgene temporarily in the plant cells. The transgene(s) thus cannot be inherited to the next generation [243] as these do not get transferred to the reproductive tissues omitting its passage to germline. Chen et al. [32] have reported that agroinfiltration-based platform is an effective, safe, low-cost, and scalable approach for the rapid production of recombinant proteins. Besides these considerable factors, agroinfiltration employing the use of vectors from recombinant plasmids will be performed at BSL-1 level facilities can be considered as transgenics. Keeping in view the biosafety aspects, agroinfiltration can pave towards the production of marker (transgene)-free transgenic plants which will overcome the biosafety concerns and can be efficiently opted for commercialization in future. The commercial production of recombinant protein(s) suffers from biosafety and risk assessment [244]. However, the agroinfiltration technique can potentially resolve these biosafety issues for improving the commercial sustainability for cost and availability of the recombinant protein(s) in the coming eras. Further, the consumer acceptance for these products can be better ascertained as agroinfiltration is a transgene-free approach. Also, it is identified as an efficient approach to develop the marker-free plants utilizing leaf disc agroinfiltration with a marker-free plasmid vector containing the target gene in tobacco plants hence limited biosafety concerns (https://www.isaaa.org/). It is now being widely used to produce commercially important recombinant proteins [245]. The production of low-yielding plant-produced recombinant proteins could be a hindrance to the commercial success of agroinfiltration [246] but the efficient use of optimized agroinfiltration-based factors can circumvent the limitation to increase the transformation efficiency in crop species.

## 17. Conclusions

Agroinfiltration coupled with genome editing holds promise for targeted site modification in the genomic sequence without the integration of the construct into the plant genome. The direct injection of Cas9-sgRNA complex in agroinfiltration process, encased in plasmid could edit the genes in the plant. Although CRISPR/Cas9 technology is the “Technology of choice” for editing genes as compared to TALENs and ZFNs, the combination of CRISPR/Cas9 with agroinfiltration, an *in planta* and transient approach makes it is a breakthrough technology by enhancing the feasibility, efficacy, and ultimately, the consumer acceptability of this technology. Further, the technique avoids the limitation of low transformation efficiency generated during stable genetic transformation methods. Moreover, agroinfiltration-based genome editing can help broaden the scope of targeted genome modification through simple, high throughput and genotype-independent genome editing approach. Thus, the convergence of genome editing (targeted site modification) and agroinfiltration (simplest approach) can improve the ease, versatility, and efficacy of plant genome transformations to achieve commercially viable and consumer-preferred plant products.

## Figures and Tables

**Figure 1 ijms-22-10882-f001:**
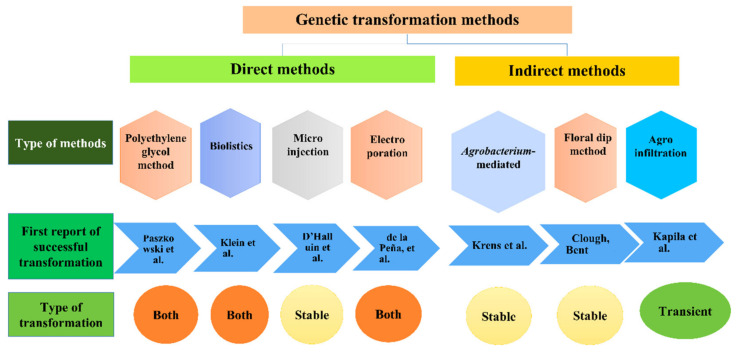
Genetic transformation methods in plants [7,8,9,10,11,12,13].

**Figure 2 ijms-22-10882-f002:**
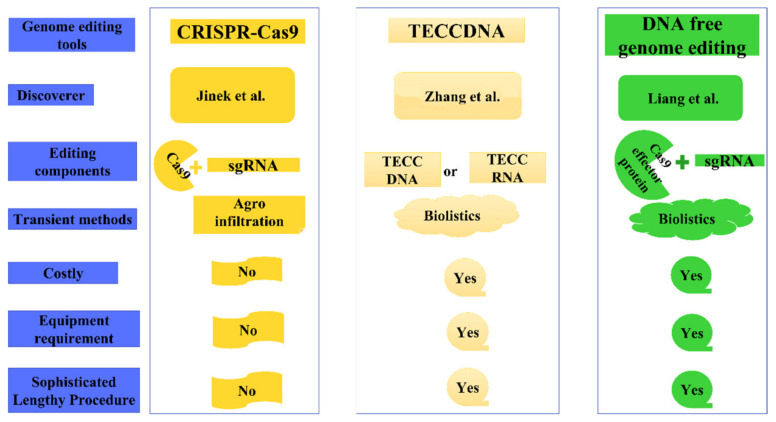
Genetic transformation methods in plants [26,27,28].

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
