# Peer review of "Agroinfiltration Mediated Scalable Transient Gene Expression in Genome Edited Crop Plants"

_ijms, 2021, doi:10.3390/ijms221910882_

Round 1

Reviewer 1 Report

Dear Authors,

Revised manuscripts have much improved the understanding of the manuscript. I am satisfied with the response to the previous comments. Hence, recommend for acceptance  after minor modification as below:

However, for all three figures, the image resolution needs to increase since the image looks blurred. Further, the image has a small font size and is not visible.

Moreover, the abstract has more than 200 words; as author guidelines abstract should be within 200 words limits.

Author Response

Dear Reviewer,

We sincerely thank you for appreciating the concept and the critical comments that have been very helpful to improve the presentation and content of the manuscript.

The point-wise reply to the comments is appended below:-

Comment 1: However, for all three figures, the image resolution needs to increase since the image looks blurred. Further, the image has a small font size and is not visible.

Reply: The images have been revised again.

Comment 2: Moreover, the abstract has more than 200 words; as author guidelines abstract should be within 200 words limits.

Reply: The length of the abstract has been reduced and now, it consists of 167 words.

warm regards

anu

Reviewer 2 Report

The review “Agroinfiltration mediated scalable transient gene expression in genome-edited crop plants“ focuses on the use of agroinfiltration in combination with gene editing and various aspects related to agroinfiltration and transient gene expression. The topic is undoubtedly relevant and interesting in connection with the development of gene editing technologies and emerging issues of biosafety. However, the review is not without its drawbacks. Basically, it is still devoted specifically to agroinfiltration as a technology, and there is not much information about the combination of agroinfiltration and gene editing. The boundaries of using agroinfiltration for editing are practically not marked. It is not discussed whether it is possible to obtain a plant with an edited genome using agroinfiltration, or agroinfiltration can only be used to test and study the functions of certain genes. The advantages of agroinfiltration over other methods for editing are not clearly indicated.

Minor points.

  1. Lines 67 – 69. Not a very correct statement. Cas9 and nRNA are not used instead of T-DNA, they are actually part of T-DNA, that is, the DNA that is contained in the plasmid between two border repeats and is transferred into the plant cell during agrobacterial transformation.
  2. Line 77. Incorrect – bombardment is not always targeted to the germinal cells. Biolistic delivery of editing tools usually requires the same procedures to produce edited plants as agrobacterial transformation (except for the floral dip method).
  3. Lines 104-106. It is not very clear what is involved in sampling herbicide-resistant or antibiotic-resistant samples? Or are antibiotics added to the medium to eliminate Agrobacterium? At the same time, one cannot fail to foresee the fact that avoiding selection on antibiotics inevitably leads to the selection of editing events, which, without a selective agent, can be quite laborious.
  4. Figure 2. Figure 2 is little discussed in the text, its role is not very clear. In addition, in the second column, it is not very clear in what form Cas9 is delivered, while the second and third columns indicate that it is delivered in the form of DNA or as a protein.
  5. Lines 133-136. It looks like a repeat of lines 64-66.
  6. Lines 162-165. Also looks like a repeat. It has already been said that Agrobacterium is the most commonly used and popular vector.
  7. Lines 175-177. Also looks like a repeat.
  8. Lines 178-182. Agrobacterium lines used to obtain transgenic plants do not cause the formation of crown galls. This paragraph is worth rewriting, and perhaps it would have looked more appropriate at the beginning of this section, which deals with the history of the development of technology.
  9. Line 188. Apparently, there is a mistake in the text, otherwise, it turns out that T-DNA delivers itself.
  10. Lines 205. Maybe “leading” instead of “led”?
  11. Lines 210-213. It is worth mentioning that one of the problems with agrobacterial transformation is that transformation and regeneration protocols have not been developed for all plant species.
  12. Line 241. Murashige and Skoog medium, not solution.
  13. Lines 261-267. Perhaps it is better to add years or other indications of the time of publication of works since this paragraph describes the development of technology.
  14. Line 281. Literary references are not formatted properly.
  15. Lines 352-354. This sentence talks about variety, but only two plant species are mentioned.
  16. Line 368. The reference is needed (in place of [77]?).
  17. Lines 461-462. A repeat of lines 143-145.
  18. Lines 461-472. Section 12 is about CRISPR, it is not very clear why this paragraph focuses so much on other editing tools. It might be worth moving, as it interrupts the logically ongoing story of using agroinfiltration to test editing tools.
  19. Lines 472-473. Sounds like all researchers are mainly concerned with knocking out the PDS gene, perhaps something else is meant?
  20. Lines 475 -476 and 477-478. These two sentences look like they are talking about the same thing in different words.
  21. Line 533 and line 589. The word “Introduction” should be removed.

Author Response

The review “Agroinfiltration mediated scalable transient gene expression in genome-edited crop plants“ focuses on the use of agroinfiltration in combination with gene editing and various aspects related to agroinfiltration and transient gene expression. The topic is undoubtedly relevant and interesting in connection with the development of gene editing technologies and emerging issues of biosafety. However, the review is not without its drawbacks.

Comment 1. Basically, it is still devoted specifically to agroinfiltration as a technology, and there is not much information about the combination of agroinfiltration and gene editing.

Reply: We sincerely thank you for sharing the critical observations. We would like to draw your attention to the following sections of the manuscript:-

  1. The combinatorial approach of agroinfiltration transient approach with genome editing serve as a pilot experiment before proceeding the stable transformation is mentioned in line 325-326 and 444-447.
  2. The manuscript contains a total of 20 references that are associated with agroinfiltration based genome editing in numerous plant species as summarized in in the Table 2. The specific references linked with agroinfiltration and genome editing are as follows: 77, 131, 156, 158,159, 160, 161, 162, 163, 164, 165, 166, 167, 168, 169, 171, 173, 175, 176, 177.

Comment 2. The boundaries of using agroinfiltration for editing are practically not marked. It is not discussed whether it is possible to obtain a plant with an edited genome using agroinfiltration, or agroinfiltration can only be used to test and study the functions of certain genes.

Reply: Agroinfiltration-based genome editing has the possibility to obtain plant with edited genome through targeting specific genes, for example,

  • Meiosis genes e SPO11-1, REC8, and OSD1 in cowpea and knock out displayed the defects in 100% of examined male and female meiocytes (Juranic et al 2020)
  • Tomato MADS box gene6 in the strawberry and phenotypic characterization of mutant lines deciphered its role in anther development (Martín-Pizarro et al 2019)
  • PDS gene in Yam and its knock out showed the disruption of its function leading to albinism (Syombua et al 2020)

These details are mentioned in Table 2 and details has been added in line 505-509.

We want to clarify that the agroinfiltration based genome editing can be used for both testing the genes and to study the functions of genes.

  1. The main feature of agroinfiltration include experiment simplicity, ease of transformation and high reproducibility (Sharma et al 2020). These features make this transient approach a direct, native and reliable system to test the single guide RNA expression and large genome editing constructs efficiency within 3 days (Sharma et al 2020, Lee and Yang 2006). After testing the construct functionality through agroinfiltration based genome editing, it can be used for stable transformation experiments (Sharma et al 2020).

Agroinfiltration is mainly used to test the transient genome editing as mentioned in line 449-451.

  1. Agroinfiltration based genome editing confirmed the functions of specific genes viz.

  • Meiosis genes e SPO11-1, REC8, and OSD1 in cowpea identified that induce mitosis from meiosis (Juranic et al 2020)
  • Tomato MADS box gene6 in strawberry has vital role in anther development (Martín-Pizarro et al 2019)
  • PDS gene in Yam is involved in carotenoid biosynthesis (Syombua et al 2020)

These details are mentioned in Table 2, line 505-508, line 553-554 and line 472-473.

Comment 3: The advantages of agroinfiltration over other methods for editing are not clearly indicated.

Reply: The agroinfiltration based CRISPR-Cas genome editing is a cost effective approach and has no requirements of equipments and sophisticated lengthy procedure as compared to Transiently Expressed CRISPR/Cas DNA (TECC DNA) and DNA-free genome editing (using ribonucleoproteins) as mentioned in Figure 2.

Comment 4: Lines 67 – 69. Not a very correct statement. Cas9 and nRNA are not used instead of T-DNA, they are actually part of T-DNA, that is, the DNA that is contained in the plasmid between two border repeats and is transferred into the plant cell during agrobacterial transformation.

Reply: The word ‘place of’ replaced with ‘the’ in the line 69, hence, the line has been modified, making it a correct statement.

Comment 5: Line 77. Incorrect – bombardment is not always targeted to the germinal cells. Biolistic delivery of editing tools usually requires the same procedures to produce edited plants as agrobacterial transformation (except for the floral dip method).

Reply: We agree with the reviewer comment. Therefore, the line 77 has been modified in the manuscript.

Comment 6: Lines 104-106. It is not very clear what is involved in sampling herbicide-resistant or antibiotic-resistant samples? Or are antibiotics added to the medium to eliminate Agrobacterium? At the same time, one cannot fail to foresee the fact that avoiding selection on antibiotics inevitably leads to the selection of editing events, which, without a selective agent, can be quite laborious.

Reply: Yes, we agree with the comments of the learned reviewer. The content has been modified for improved clarity.

The text ‘Generally the plant genome editing tools require additional cycles of plant regeneration under antibiotic selection medium but TECCDNA approach excluded the addition of herbicide or antibiotics addition to the medium meant for the selection of transformed plants. After biolistic delivery of TECCDNA, embryos were transferred for callus induction, regeneration on rooting medium that generated large number of T0 seedlings about 1 week later provided no selective agents were used during tissue culture process.’ has been added to line 106.

Comment 7: Figure 2. Figure 2 is little discussed in the text, its role is not very clear. In addition, in the second column, it is not very clear in what form Cas9 is delivered, while the second and third columns indicate that it is delivered in the form of DNA or as a protein.

Reply: The Figure 2 discussion in the text is enhanced.

Cas9 delivered in three ways; DNA, RNA and protein (Yip 2020). In Figure 2, first column Cas9 is delivered in the form of DNA, while in the 2nd column- Cas9 is delivered in the form of DNA or RNA and in 3rd column- Cas9 is delivered in the form of protein.

Yip B. H. Recent Advances in CRISPR/Cas9 Delivery Strategies. Biomolecules2020, 10(6), 839. https://doi.org/10.3390/biom10060839

Comment 8: Lines 133-136. It looks like a repeat of lines 64-66.

Reply: Yes, the lines 134-136 have been deleted from the manuscript.

Comment 9: Lines 162-165. Also looks like a repeat. It has already been said that Agrobacterium is the most commonly used and popular vector.

Reply: Yes, the lines 162-165 have been deleted in the revised manuscript. The reference [47] present in the line 162 shifted to line 161.

Comment 10: Lines 175-177. Also looks like a repeat.

Reply: Lines 175-177 have been deleted in the revised manuscript. The phrase ‘for plant modifications’ has been added and the references [64] and [65] cited in the line 173.

Comment 11: Lines 178-182. Agrobacterium lines used to obtain transgenic plants do not cause the formation of crown galls. This paragraph is worth rewriting, and perhaps it would have looked more appropriate at the beginning of this section, which deals with the history of the development of technology.

Reply: The modified text used in line 178-182 has been shifted to line 142 that displayed the history of Agrobacterium based genetic transformation.

Comment 12: Line 188. Apparently, there is a mistake in the text, otherwise, it turns out that T-DNA delivers itself.

Reply: We agree with the reviewer comments. The indicated text ‘T-DNA can help in transfer of T-DNA through” has been replaced with “-based genetic transformation undergo” in line 188.

Comment 13: Lines 205. Maybe “leading” instead of “led”?

Reply: The word ‘led’ replaced with ‘leading’.

Comment 14: Lines 210-213. It is worth mentioning that one of the problems with agrobacterial transformation is that transformation and regeneration protocols have not been developed for all plant species.

Reply: The text ‘thus regeneration protocols have not been developed for all plant species.’ Has been added in the line 214.

Agrobacterium-mediated approaches have been employed to almost 85% of all species (Wu and Zhao, 2017, Paes de Melo et al. 2020).

Wu, E.; Zhao, Z.-Y. Agrobacterium-mediated sorghum transformation,” in plant germline development methods in Molecular Biology. Ed. Schmidt, A. (New York, NY: Springer New York), 2017, 355–364. doi: 10.1007/978-1-4939-7286-9_26

Paes de Melo, B.; Lourenço-Tessutti, I.T.; Morgante, C.V.;Santos,N.C.; Pinheiro, L.B.; de Jesus Lins, C.B.; Silva, M.C.M; Macedo, L.L.P.; Fontes, E.P.B.; Grossi-de-Sa,M.F. Soybean embryonic axis transformation: combining biolistic and Agrobacterium-mediated protocols to overcome typical complications of in vitro plant regeneration. Front. Plant Sci. 2020, 11: 1228. https://doi.org/10.3389/fpls.2020.01228

Comment 15:  Line 241. Murashige and Skoog medium, not solution.

Reply: The word ‘solution’ has been replaced with ‘medium’ in line 241.

Comment 16: Lines 261-267. Perhaps it is better to add years or other indications of the time of publication of works since this paragraph describes the development of technology.

Reply: The year with respect to the FAST technology has been added in the line 262 and the year corresponding to the AGROBEST technology has been added in the line 267.

Comment 17: Line 281. Literary references are not formatted properly.

Reply: Literary references has been formatted properly from (21,92,114) to [21,92,114] in line 281.

Comment 18: Lines 352-354. This sentence talks about variety, but only two plant species are mentioned.

Reply: The text ‘; Nicotiana benthamiana [79,126] and Spinacia oleracea [78]’ has been replaced with’ (Table 1-3) in line 354 of the revised manuscript.

Comment 19: Line 368. The reference is needed (in place of [77]?).

Reply: The pertinent reference has been incorporated at the end of the sentence.

Comment 20: Lines 461-462. A repeat of lines 143-145.

Reply: Lines 461-462 have been deleted in the revised manuscript to avoid repetition.

Comment 21: Lines 461-472. Section 12 is about CRISPR, it is not very clear why this paragraph focuses so much on other editing tools. It might be worth moving, as it interrupts the logically ongoing story of using agroinfiltration to test editing tools.

Reply: The text has been modified from line 461- 472 to improve the readability and flow of contents in the revised manuscript.

Comment 22: Lines 472-473. Sounds like all researchers are mainly concerned with knocking out the PDS gene, perhaps something else is meant?

Reply: Yes, we meant that the researchers targeted knocking out of the PDS gene as provided in a total of 11 references i.e 156, 158, 159, 160, 161, 162, 163, 164, 165, 166, 167 which have been cited in the Table 2. These are associated with editing of PDS gene using agroinfiltration. However, the researchers targeted 11 genes other than PDS gene that are associated with agroinfiltration based genome editing i.e

  1. Green fluorescent Protein gene
  2. Isopentenyl/dimethylallyldiphosphate synthase genes
  3. Six sites of Bean yellow dwarf

virus genome

  1. Xylosyl transferase gene
  2. Proliferating cell nuclear antigen gene
  3. Immunity associated gene
  4. Canker susceptibility gene
  5. Tomato MADS box gene6
  6. 3′-hydroxyl-N-methylcoclaurine 4′-O-methyltransferase gene
  7. Meiosis genes
  8. Centromere-specific histone H3

These details are mentioned in Table 2.

Comment 23: Lines 475 -476 and 477-478. These two sentences look like they are talking about the same thing in different words.

Reply: The lines 475-476 have been deleted as per the suggestion.

Comment 24: Line 533 and line 589. The word “Introduction” should be removed.

Reply: The word ‘Introduction’ has been removed from the manuscript in line 533 and line 589.

Reviewer 3 Report

Review of the current status of genetic editing in plants by Agroinfiltration .  Approaches are explained and their combination with other gene-editing technologies such as CRISPR-Cas9 are explored.  Comparison is made with existing standard approaches to plant genome editing with agrobacterium and T-DNA. This is a useful update on the current status of gene editing approaches.  Stylistically, some reduction in the length of sentences would assist in making the manuscript more readable.  For example, lines 33-36 in the abstract, 75-80.   

Author Response

Comment 1: Review of the current status of genetic editing in plants by Agroinfiltration.  Approaches are explained and their combination with other gene-editing technologies such as CRISPR-Cas9 are explored.  Comparison is made with existing standard approaches to plant genome editing with agrobacterium and T-DNA. This is a useful update on the current status of gene editing approaches.

Reply: We sincerely thank you for appreciating the revised manuscript. 

Comment 2: Stylistically, some reduction in the length of sentences would assist in making the manuscript more readable.  For example, lines 33-36 in the abstract, 75-80.

Reply: The abstract has been modified to streamline the readability. The word count of the revised abstract is reduced (from 239 words of the original abstract to 167 words).

Lines 33-36 have been modified in the revised manuscript.

Line 75-80 has been modified in the revised manuscript.

This manuscript is a resubmission of an earlier submission. The following is a list of the peer review reports and author responses from that submission.

Round 1

Reviewer 1 Report

The review article “Agroinfiltration- Ushering the functional transient genetic transformation for improved genome editing efficiencies in crop plants” submitted by Kaur et al. summarizes agroinfiltration related research conducted in different plant species. I see a fundamental issue in this review as the authors propose that the agroinfiltration could be an alternative method to stable transformation for producing gene edited crops.

The following points should explain why Agroinfiltration is not a suitable method:

  1. Agroinfiltrations are performed mostly on somatic tissues such as leaves, which do not transmit the transgene to next generation. Transgene expression is mainly observed in the infiltrated region of the leaf, and it does not even transmit the expressed transgene protein to the adjacent area. Many R-gene researchers take advantage of this phenomenon to display the reaction of multiple R gene types on a single tobacco leaf through agroinfiltration.
  2. Agroinfiltration was used to test the efficiency of gRNA as the infiltrated region will efficiently express the Cas9 and gRNA. The cells will have different types of edits and sequencing the target regions will show the editing efficiency. However, these edits are not useful for functional studies as 1) there will be different types of edits in homozygous and heterozygous conditions in each cell 2) the edits will not produce phenotypes such as plant height, flowering time, root length, and fruit color, etc. (genes like PDS are exceptions) 3) the alleles produced through gene editing cannot be saved as these are not transmitted to next generation.
  3. In planta transformation and bombarding the protein-RNA complexes target the germinal cells, where the editing will be transferred to next generation. This will not be the case in Agroinfiltrations.
  4. Regarding the biosafety issue, the USA has already approved gene editing through Agrobacterium mediated transformations. Back crosses can be performed to remove the CRISPR-Cas9 vector insertion and retain only the beneficial gene edited alleles. Additionally, Agroinfiltrations also use vectors from recombinant plasmids, these experiments will be performed at BSL-1 level facilities and probably considered as transgenics.
  5. Almost all the studies mentioned in Table 2. ‘Agroinfiltration based genome editing strategies for plants’, used Agroinfiltration to test the efficiency of the gRNA and not to for gene function studies. Only in the opium poppy study, they used Agroinfiltrations to mutate a gene in a metabolite pathway in leaves. There were lots of mistakes in Table 2, for example, reference #168 for the second study on Arabidopsis is not the correct one.

Reference #77 the original study did not include tomato.

Reference # 160 and 176, only gRNAs were tested in sweet orange and purple cabbage using Agroinfiltration, respectively.

Reference # 173, First gRNA was tested in Agroinfiltration and stable transgenics were produced for gene characterization. Similarly, it was the same case for reference #177 strawberry, reference # 166 yam, and reference 131 cowpea studies.

Author Response

Dear Reviewer,

We sincerely thank for your critical observations and suggestion.

We have tried our best to reply back to your indicated observations and also have incorporated the changes in the revised manuscript.

Here are the point-wise replies to your comments:-

Reviewer 1:

Comments and Suggestions for Authors

The review article “Agroinfiltration- Ushering the functional transient genetic transformation for improved genome editing efficiencies in crop plants” submitted by Kaur et al. summarizes agroinfiltration related research conducted in different plant species. I see a fundamental issue in this review as the authors propose that the agroinfiltration could be an alternative method to stable transformation for producing gene edited crops.

Answer: We thank the learned reviewer for the useful comments. The fundamental utility of the agroinfiltration technique is its transient transformation capabilities.

 The following points should explain why Agroinfiltration is not a suitable method:

Comment 1a: Agroinfiltrations are performed mostly on somatic tissues such as leaves, which do not transmit the transgene to next generation. Transgene expression is mainly observed in the infiltrated region of the leaf, and it does not even transmit the expressed transgene protein to the adjacent area.

Answer: We agree to the comments of the reviewer. Agroinfiltration process is a transient process which has been mostly tested or evaluated for the somatic cells. Actually, it forms a part of the initial screening to identify whether a particular plant tissue type is amenable to transformation or not which is indirectly validated through the sgRNA expression (genome editing) or genetic transformation (gfp gene or other gene studies). The same has been elaborated in the manuscript under 6th heading i.e. Transient (temporary gene expression).

Comment 1b: Many R-gene researchers take advantage of this phenomenon to display the reaction of multiple R gene types on a single tobacco leaf through agroinfiltration.

Answer: Regarding the R gene researches, here are a few pertinent reports. Van der Hoorn et al 2000 first reported the transient expression of resistance genes Cf-9 and Cf-4 of tomato plants through agroinfiltration in tobacco. Ma et al (2012) also observed that co-expression of R gene with the corresponding Avr gene triggering host-defence responses in a hypersensitive response in Nicotiana tabacum. They reported the formation of necrotic sector through the expression of both genes in the overlapping region that led to the transient production of R and Avr proteins in the infiltrated leaf area through agroinfiltration.

Van der Hoorn, R.A.L.; Laurent, F.; Roth, R.; De Wit, P.J.G.M. Agroinfiltration is a versatile tool that facilitates comparative analyses of Avr9/Cf-9-induced and Avr4/Cf-4-induced necrosis. Mol. Plant-Microbe Interact. 2000, 13, 439–446.

Ma, J.; Xiang, H.; Donnelly, D.J.; Meng, F.-R.; Xu, H.; Durnford, D.; Li, X.-Q. Genome editing in potato plants by agrobacterium-mediated transient expression of transcription activator-like effector nucleases. Plant Biotechnol. Rep. 2017, 11, 249–258.

It has already been present in the manuscript under [144] reference number.

Comment 2: Agroinfiltration was used to test the efficiency of gRNA as the infiltrated region will efficiently express the Cas9 and gRNA. The cells will have different types of edits and sequencing the target regions will show the editing efficiency.

Answer: The combined approach of CRISPR-Cas9 and agroinfiltration has wide application to test the efficiency of sgRNA in genome editing constructs. It has been evident that the predicted sgRNAs often have different editing efficiencies. Moreover, in silico prediction tools have been utilized to analyze the editing efficiency based on the choice and sequence feature of the sgRNA (Sharma et al 2020). Transient expression is basically an approach for the verification of transformation construct activity and for the validation of formation of small amounts of recombinant protein (Bhatia et al 2015).

Sharma, R.; Liang, Y.; Lee, M.Y.; Pidatala, V.R.; Mortimer, J.C.; Scheller, H.V. Agrobacterium‑mediated transient transformation of sorghum leaves for accelerating functional genomics and genome editing studies. BMC Res. Notes. 2020, 13, 116.

It has already been present in the manuscript under [169] reference number.

Bhatia, S,; Sharma, K.; Dahiya, R.; Bera, T. Modern applications of plant biotechnology in pharmaceutical sciences, 1st Edition, Academic Press, Cambridge, 2015.

Comment 3: However, these edits are not useful for functional studies as

Comment 3a: There will be different types of edits in homozygous and heterozygous conditions in each cell

Answer: Yes, we agree that the predicted sgRNAs often exhibit different editing efficiencies. Moreover, in silico prediction tools can help analyze the editing efficiency based on the choice and sequence feature of sgRNA (Sharma et al 2020). Transient expression is basically an approach for the verification of transformation construct activity and for the validation of formation of small amounts of recombinant protein (Bhatia et al 2015).

Sharma, R.; Liang, Y.; Lee, M.Y.; Pidatala, V.R.; Mortimer, J.C.; Scheller, H.V. Agrobacterium‑mediated transient transformation of sorghum leaves for accelerating functional genomics and genome editing studies. BMC Res. Notes. 2020, 13, 116.

It has already been present in the manuscript under [169] reference number.

Bhatia, S,; Sharma, K.; Dahiya, R.; Bera, T. Modern applications of plant biotechnology in pharmaceutical sciences, 1st Edition, Academic Press, Cambridge, 2015.

Comment 3b: The edits will not produce phenotypes such as plant height, flowering time, root length, and fruit color, etc. (genes like PDS are exceptions)

Answer: The repercussions of the genome edits under in vivo conditions are always subtle and not easily detectable. Also, the variation in the knock-outs or sequences in any of the genes after genome editing events will be more useful to study the impact of genome editing or its efficiency rather than obtaining variations in the phenotypic characters.

Agroinfiltration tend to show bleached patches as a phenotypic marker (through knockout of phytoene desaturase gene) that can be analyzed through microscopic examination of an infiltrated leaf section, Moreover, if GFP gene is taken as marker, then it can be analyzed through fluorescence microscopy of the infiltrated section (Syombua et al 2020). Later on, the transient expression can be followed by stable genetic transformation that tend to show phenotype such as dwarf phenotype (Arora and Narula 2017), albino shoot with a bushy phenotype and variegated albino plantlets (Syombua et al 2020). Also, the variation in the sequence can be easily identified through Sanger’s sequencing technique under in vitro conditions. Further,

Syombua, E.D.; Zhang, Z.; Tripathi, J.N. A CRISPR/Cas9-based genome-editing system for yam (Dioscorea spp.). Plant Biotechnol. J. 2020, 1–3.

It has already been present in the manuscript under [166] reference number.

Arora, L.; Narula, A. Gene editing and crop improvement using CRISPR-Cas9 system. Front. Plant Sci. 2017, 8. doi:10.3389/fpls.2017.01932

Comment 3c: The alleles produced through gene editing cannot be saved as these are not transmitted to next generation.

Answer: The scenario for agroinfiltration is transient model and thus exhibits the non-transmissible mode. It can be further suggested that agroinfiltration can be the first line of experiments to test the extent of transient genome editing before initiating the elaborate genome editing experiments.

It has been reported that inheritable mutations adhering to Mendel's law only have the potential to get transferred to the next generations (Osakabe and Osakabe 2017). But this is not the case for agroinfiltration because transgenes are transiently expressed in somatic cells of plant tissues through this approach and not germ cells hence not heritable in nature (Debler et al 2021).

Osakabe, Y.; Osakabe, K. Genome editing to improve abiotic stress responses in plants. Prog. Mol. Biol. Transl. Sci. 2017, 149, 99–109.

It has already been present in the manuscript under [157] reference number.

Debler, J.W.; Henares, B.M.; Lee, R.C. Agroinfiltration for transient gene expression and characterisation of fungal pathogen effectors in cool-season grain legume hosts. Plant Cell Rep. 2021, 40, 805–818. https://doi.org/10.1007/s00299-021-02671-y.

Comment 4: In planta transformation and bombarding the protein-RNA complexes target the germinal cells, where the editing will be transferred to next generation. This will not be the case in Agroinfiltrations.

Answer: We agree to the suggestion of the learned reviewer. This sentence has been modified in the revised manuscript.

Agroinfiltration depends on the transient transformation of somatic plant cells that lead to the production of transgene-encoded protein in the transformed cells (Debler et al 2021). The advent of targeting the somatic cells i.e. leaves does not allow the foreign gene to be transferred in the next generation, hence, no chance of transgene to be transferred towards the next generation. Moreover, somatic cells in the zone of agroinfiltration have the potential to express the transgenes under the control of the constitutive CaMV-35S promoter within 3-5 days after agroinfiltration (Debler et al 2021).

Debler, J.W.; Henares, B.M.; Lee, R.C. Agroinfiltration for transient gene expression and characterisation of fungal pathogen effectors in cool-season grain legume hosts. Plant Cell Rep. 2021, 40, 805–818. https://doi.org/10.1007/s00299-021-02671-y.

Comment 5: Regarding the biosafety issue, the USA has already approved gene editing through Agrobacterium mediated transformations. Back crosses can be performed to remove the CRISPR-Cas9 vector insertion and retain only the beneficial gene edited alleles. Additionally, Agroinfiltrations also use vectors from recombinant plasmids, these experiments will be performed at BSL-1 level facilities and probably considered as transgenics.

Answer: Yes, we agree with the observations that back crosses can be performed to get rid of the vector insertions. What we want to emphasize here is that agroinfiltration can be performed as a preliminary screening technique for somatic or germinal cells and later the stable transformation technique can be followed.

Regarding the possibility of considering agroinfiltration to lead to development of transgenics, following excerpts can be helpful to substantiate the opposite:-

Chen et al (2013) have reported that agroinfiltration-based platform is an effective, safe, low-cost and scalable approach for the rapid production of recombinant proteins. Also, it is identified as an efficient approach to develop the marker-free plants utilizing leaf disc agroinfiltration with marker-free plasmid vector containing the target gene in tobacco plants hence limited biosafety concerns (https://www.isaaa.org/).

Chen, Q.; Lai, H.; Hurtado, J.; Stahnke, J.; Leuzinger, K.; Dent, M. Agroinfiltration as an effective and scalable strategy of gene delivery for production of pharmaceutical proteins. Adv. Tech. Biol. Med. 2013, 1, 103.

It has already been present in the manuscript under [23] reference number.

Comment 6: Almost all the studies mentioned in Table 2. ‘Agroinfiltration based genome editing strategies for plants’, used Agroinfiltration to test the efficiency of the gRNA and not for gene function studies. Only in the opium poppy study, they used Agroinfiltrations to mutate a gene in a metabolite pathway in leaves.

Answer: Yes, we agree to the observation of the reviewer. The transient agroinfiltration approach is widely employed for the prediction of gRNA efficiency for CRISPR editing in vivo (Table 2) (Liang et al 2019). The combinatorial approach of CRISPR-Cas editing and agroinfiltration led to generation of the transformed tissue within 3 days after infiltration that serve as a reliable assay for testing sgRNAs under native conditions (Sharma et al 2020).

Liang, Y.; Eudes, A.; Yogiswara, S.; Jing, B.; Benites, V.T.; Yamanaka, R.; Cheng-Yue, C.; Baidoo, E.E.; Mortimer, J.C.; Scheller, H.V.; Loqué, D. A screening method to identify efficient sgRNAs in Arabidopsis, used in conjunction with cell-specific lignin reduction. Biotechnol. Biofuels.  2019, 12, 130.

Sharma, R.; Liang, Y.; Lee, M.Y.; Pidatala, V.R.; Mortimer, J.C.; Scheller, H.V. Agrobacterium‑mediated transient transformation of sorghum leaves for accelerating functional genomics and genome editing studies. BMC Res. Notes. 2020, 13, 116.

It has already been present in the manuscript under [169] reference number.

Comment 7: There were lots of mistakes in Table 2, for example, reference #168 for the second study on Arabidopsis is not the correct one.

Answer: The indicated changes have been incorporated in the revised manuscript.

Jiang, W.; Zhou, H.; Bi, H.; Fromm, M.; Yang, B.; Weeks, D. P. Demonstration of CRISPR/Cas9/sgRNA-mediated targeted gene modification in Arabidopsis, tobacco, sorghum and rice. Nucleic Acids Res. 2013, 41(20), e188–e188. doi:10.1093/nar/gkt780

Comment 8: Reference #77 the original study did not include tomato.

Answer: As indicated the appropriate reference has been incorporated in the revised manuscript.

Zhang, N.; Roberts, H.M.; Eck, J.V.; Martin, G.B. Generation and molecular characterization of CRISPR/Cas9-induced mutations in 63 immunity-associated genes in tomato reveals specificity and a range of gene modifications. Front. Plant Sci. 2020. https://doi.org/10.3389/fpls.2020.00010

Comment 9: Reference # 160 and 176, only gRNAs were tested in sweet orange and purple cabbage using Agroinfiltration, respectively. Reference # 173, First gRNA was tested in Agroinfiltration and stable transgenics were produced for gene characterization. Similarly, it was the same case for reference #177 strawberry, reference # 166 yam, and reference 131 cowpea studies.

Answer: Yes, we thoroughly agree with the observation and comment of the reviewer. These references do signify the sgRNA efficiency in orange and purple cabbage using Agroinfiltration [160,176]. Likewise, agroinfiltration transient approach has been tested followed by implementation through stable transformation for gene characterization studies in strawberry [177], yam [166] and cowpea [131].

Thanks again.

best regards

Reviewer 2 Report

Dear Authors,

This review manuscript regarding Agroinfiltration/ transient transformation in plants is well written and covers all previous study aspects. However, the manuscript has some flaws in presentation and must be corrected to improve the understanding of the manuscript. Hence, recommend for minor revision:

Figure 1, Figure 2 and 3: In all figures, the resolution of an image needs to increase. Further, these images have small font sizes and not properly visible. Make font size visible.

In the reference section: Check the references style according to the journal format.

In reference 176: Brassica oleracea should be italic.

In reference 246: Arabidopsis thaliana should be italic.

Author Response

Dear Reviewer,

Sincere thanks for approving to the contents of the manuscript.

We have attended the indicated queries and incorporated the changes accordingly in the revised manuscript.

Here are the point-wise replies:-

Reviewer 2:

Comments and Suggestions for Authors

Dear Authors,

This review manuscript regarding Agroinfiltration/ transient transformation in plants is well written and covers all previous study aspects. However, the manuscript has some flaws in presentation and must be corrected to improve the understanding of the manuscript. Hence, recommend for minor revision:

Comment 1: Figure 1, Figure 2 and 3: In all figures, the resolution of an image needs to increase. Further, these images have small font sizes and not properly visible. Make font size visible.

Answer: All the figures have been replaced with improved resolution images in the revised manuscript. The font size has also been increased as suggested.

Comment 2: In the reference section: Check the references style according to the journal format.

Answer: The reference section has been revised.

Comment 3: In reference 176: Brassica oleracea should be italic.

Answer: The name of the crop has been italicized in the revised manuscript.

Comment 4: In reference 246: Arabidopsis thaliana should be italic.

Answer: The name has been italicized as suggested in the revised manuscript.

Sincere thanks and warm regards

Round 2

Reviewer 1 Report

This review proposes that the agroinfiltrations could be an alternate for stable transformation in gene editing studies, but without proper supporting studies. In the first review, I pointed out how the Gene editing using Agroinfiltration, so far,  was mostly used for testing the gRNA efficiency and not for gene functional studies. The author's response for the comments did not address any of the major concerns raised in the first review.